# High-resolution line-scan Brillouin microscopy for live imaging of mechanical properties during embryo development

Carlo Bevilacqua [1,2], Juan Manuel Gomez[3], Ulla-Maj Fiuza [1,9], Chii Jou Chan[1,10,11], Ling Wang [1], Sebastian Hambura [1], Manuel Eguren[1], Jan Ellenberg [1], Alba Diz-Muñoz [1], Maria Leptin[3] & Robert Prevedel [1,4,5,6,7,8] ✉

Brillouin microscopy can assess mechanical properties of biological samples in a three-dimensional (3D), all-optical and hence non-contact fashion, but its weak signals often lead to long imaging times and require an illumination dosage harmful for living organisms. Here, we present a high-resolution line-scanning Brillouin microscope for multiplexed and hence fast 3D imaging of dynamic biological processes with low phototoxicity. The improved background suppression and resolution, in combination with fluorescence light-sheet imaging, enables the visualization of the mechanical properties of cells and tissues over space and time in living organism models such as fruit flies, ascidians and mouse embryos.

The fields of mechanobiology and physical biology have grown quickly in the past decade, revealing that cell and tissue mechanics play an integral role in determining biological function[1]. Although molecular components of cells can routinely be visualized with fluorescence microscopy in biology, assessment of the mechanical, that is, elastic and viscous, properties of living cells with similar spatiotemporal resolution is more challenging given the intrinsic limitations of current biophysical techniques[2–4]. Recently, Brillouin spectroscopy has emerged as a non-destructive, label- and contact-free technique that can assess the viscoelastic properties of biological samples via photon–phonon scattering interactions[5–7]. By analyzing the frequency shift ($v_B$) and linewidth ($\Gamma_B$) of the light that is inelastically scattered from gigahertz-frequency longitudinal acoustic vibrations (phonons) in the sample, one can deduce the elastic and viscous properties of the sample, respectively. When coupled to a confocal microscope[8], Brillouin microscopy can achieve three-dimensional (3D) diffraction-limited resolution. Brillouin

microscopy has enabled a wide range of applications in biology, including studies of intracellular biomechanics in living cells[9], of ex vivo[10,11] and in vivo tissues[12–14] and their components[15] (for example, collagen and elastin, as well as biomaterials[16]), and early disease diagnosis[17]. However, despite its unique advantages, many 3D or live-imaging applications in biology remain practically out of reach due to the fast dynamics or the photosensitivity of the biological samples involved. This is because virtually all Brillouin microscopy implementations are based on the single-point scanning (confocal) principle. This, together with the weak Brillouin scattering signal, typically leads to 2D imaging times in excess of tens of minutes to hours for large samples. At the same time, the confocal approach entails high and potentially phototoxic illumination dosages. Although the improved signal levels of stimulated Brillouin scattering modalities[18,19] have the potential to decrease effective image acquisition time, the required high illumination dosages (~260 mW, ref. [18]) limit their use for live imaging of sensitive samples over extended time periods.

[1]Cell Biology and Biophysics Unit, European Molecular Biology Laboratory, Heidelberg, Germany. [2]Collaboration for joint PhD degree between EMBL and Heidelberg University, Faculty of Biosciences, Heidelberg University, Heidelberg, Germany. [3]Director's Research Unit, European Molecular Biology Laboratory, Heidelberg, Germany. [4]Developmental Biology Unit, European Molecular Biology Laboratory, Heidelberg, Germany. [5]Epigenetics and Neurobiology Unit, European Molecular Biology Laboratory, Monterotondo, Italy. [6]Molecular Medicine Partnership Unit (MMPU), European Molecular Biology Laboratory, Heidelberg, Germany. [7]German Center for Lung Research (DZL), Heidelberg, Germany. [8]Interdisciplinary Center of Neurosciences, Heidelberg University, Heidelberg, Germany. [9]Present address: Systems Bioengineering, MELIS, Universidad Pompeu Fabra, Barcelona, Spain. [10]Present address: Mechanobiology Institute, National University of Singapore, Singapore, Singapore. [11]Present address: Department of Biological Sciences, National University of Singapore, Singapore, Singapore. ✉e-mail: prevedel@embl.de

Here, we present a Brillouin microscope specifically designed to address the limitation mentioned above and to enable a wider range of applications in cell and developmental biology. Our microscope is based on a line-scanning approach that enables multiplexed signal acquisition, allowing for the simultaneous sensing of hundreds of points and their spectra in parallel. Furthermore, the use of a dual objective system in a 90° configuration together with near-infrared illumination ensures minimal photodamage and phototoxicity. In contrast to previous proof-of-principle work by Zhang et al.[20], our system design is specifically optimized for high spatiotemporal resolution as well as high signal-to-noise ratio, low phototoxicity and physiological sample mounting to enable long-term 4D mechanical imaging of highly sensitive biological specimens with subcellular resolution. An in-built, concurrent selective plane illumination microscopy (SPIM) fluorescence imaging modality enables 3D fluorescence-guided Brillouin image analysis, and further aids in data interpretation and in correlating and assigning mechanical properties to different (molecular) constituents or tissue regions (Supplementary Note 1). Furthermore, we implemented a graphics processing unit (GPU)-optimized, fast numerical fitting routine for real-time spectral data analysis and visualization, thus providing rapid feedback to the experimenter. We demonstrate the capabilities of our line-scan Brillouin microscope (LSBM) by live-imaging the 3D mechanical properties of developing *Drosophila melanogaster*, *Phallusia mammillata* and mouse embryos over a field of view (FOV) of up to ~186 × 165 × 172 µm with spatial resolution down to 1.5 µm and temporal resolution down to 2 min.

## Results

### Line-scan Brillouin microscopy system

We designed the LSBM imaging system as shown in Fig. 1. It is based on an inverted SPIM configuration[21] with a high (0.8) numerical aperture, as well as a narrowband (50 kHz) yet tunable 780 nm diode laser (Methods and Extended Data Fig. 1). This wavelength substantially reduces phototoxic effects compared with 532 nm laser lines that are more commonly used in confocal Brillouin microscopy[8,14,22]. Moreover, the near-infrared wavelength does not interfere with the excitation spectra of common fluorescence labels. The tuneability of the laser enables the frequency to be stabilized by locking it to well-defined atomic transitions by means of absorption spectroscopy (the $D_2$ line of $^{87}Rb$) and the use of a gas cell as an ultra-narrowband notch filter[13,23] for the suppression of inelastically scattered Rayleigh light to within ~80 dB. Such high background suppression is critical for imaging of biological tissues. To ensure the necessary high spectral purity of the diode laser, we built a custom narrowband filter consisting of a Bragg grating and a cavity-stabilized Fabry–Pérot interferometer to suppress amplified spontaneous emission noise below ~90 dB (Extended Data Fig. 2 and Methods).

The optomechanical design of the microscope enables operation in two alternative geometries for Brillouin imaging (Fig. 1c,d and Extended Data Fig. 1): either the illumination and detection axis can be separated in a 90° fashion (termed 'orthogonal-line' or O-LSBM, Fig. 1c), similar to SPIM, to minimize the total illumination dosage for volumetric imaging, or the sample can be illuminated by a focused line in a 180° backscattered fashion (termed 'epi-line' or E-LSBM) to mitigate the effects of scattering and optical aberrations (Fig. 1d). In both cases, the optical parameters of the microscope and the overall performance of the spectrometer were optimized for high spatial resolution (matched to typical phonon lengths in biological material[24]) and background suppression, as well as the comparatively large FOV (~200 µm) required for common organism models in biology (Extended Data Figs. 3–5). To maintain homogenous focusing of the illumination line across an ~200 µm FOV in the O-LSBM configuration, we use an electrically tunable lens (Fig. 1e, Extended Data Fig. 6 and Methods). A detailed discussion of the performance and various trade-offs between the two modalities (O- and E-LSBM), a comparison with confocal Brillouin

microscopy, and details of the fundamental limits of the spatial and mechanical resolution of Brillouin microscopy, are summarized in Supplementary Note 1.

The microscope's sample stage incorporates a fluorinated ethylene propylene foil to physically isolate the specimen chamber from the objective's immersion media, which enables the use of microdrop cultures required for longitudinal embryo imaging. Furthermore, we designed a custom miniaturized incubation chamber for full environmental (temperature, $CO_2$, $O_2$) control (Fig. 1b and Extended Data Fig. 1c,d). To analyze the multiplexed (100+) spectra of the LSBM in real time and ensure high-precision spectral measurements, we implemented a GPU-enhanced numerical fitting routine tailored to the line shape of our experimental spectra[25], yielding a >1,000-fold enhancement in processing time and thus real-time spectral data analysis while showing accuracy comparable to the CPU-based pipeline (Extended Data Fig. 7 and Methods). Overall, the line-scanning Brillouin imaging system achieves a shot noise-limited signal-to-noise ratio (approximately >10 dB) and spectral precision (<20 MHz) that are comparable to single-point scanning Brillouin microscopy implementations[9,14,26], and thus should be suitable for the imaging of biological samples at a much reduced illumination dose (Fig. 1f and Extended Data Figs. 4d,10). Together, this represents substantially increased performance compared with earlier implementations of line-scanning Brillouin microscopy[20] (Supplementary Note 1).

### Fast Brillouin imaging during *Drosophila* gastrulation

To demonstrate the capabilities of the LSBM we captured longitudinal and 3D mechanical information of various developing organisms on physiologically relevant timescales. First, to highlight the capacity of the LSBM to detect mechanical changes of highly dynamic processes such as organismal-level morphogenesis, we recorded two fast and widely studied events in *Drosophila* gastrulation, ventral furrow formation (VFF) and posterior midgut invagination (PMI) (Fig. 2). VFF and PMI are fast tissue-folding events that occur on a timescale of minutes, which are both driven by actomyosin but which have different geometries in their contractile domain (rectangular in VFF, circular in PMI)[27] (Fig. 2a,b,e,f). During early VFF, cells with a mesodermal fate (marked by the expression of the transcription factor Snail) located on the ventral side of the embryo undergo complex changes in cell geometry that will enable the complete invagination of these cells in ~15 min[28] (Fig. 2a,b). Using E-LSBM we were able to record both fluorescence SPIM and Brillouin shift maps of VFF over an FOV of ~22 × 180 × 71 µm (z increment of 1.5 µm) and of PMI over an FOV of ~83 × 183 × 43 µm (z increment of 2.5 µm), both sampled with a volume time resolution of ~2 min (Fig. 2c,f and Supplementary Videos 1,2). This equals ~10 s per 2D image slice or an effective pixel time of ~1 ms inside live biological tissues, which represents an approximate 100-fold and 20-fold improvement compared with previous spontaneous and stimulated Brillouin scattering microscopes, respectively, at more than 10-fold lower illumination energy per pixel[7]. On analysis of the volumetric Brillouin data based on fluorescence membrane labels, we found that the average Brillouin shift of cells engaged in VFF transiently increases inside the potential mesoderm by 10–20 MHz from the point of apical constriction initiation (timepoint 0 min) until mesoderm invagination is completed (paired one-way ANOVA, $P = 0.066$; post-hoc multiple comparison test, $P = 0.039$, Fig. 2c,d and Supplementary Video 1). We then imaged PMI, in which the cells form a circular contractile domain (Fig. 2e,f and Supplementary Video 2). Similar to the observations during VFF, the average Brillouin shift inside cells engaged in tissue folding also increased during PMI (Fig. 2f). This suggests that during tissue folding a higher Brillouin shift is a common feature independent of the geometry of the contractile domain. No photodamage or phototoxicity were observed at less than ~20 mW of average laser power, and viability assays showed that all embryos ($n = 3$) that were imaged progressed to the first larval stage (24 h post-fertilization).

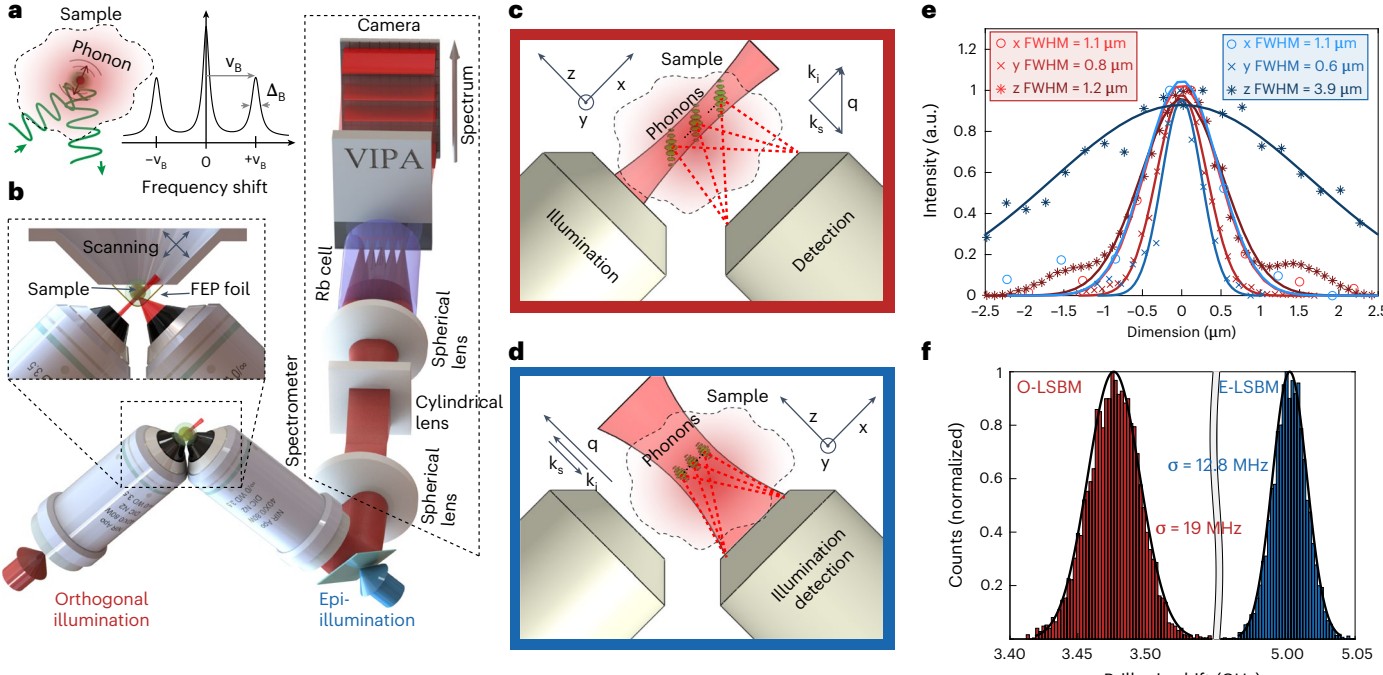

**Fig. 1 | Schematics and characterization of the LSBM. a**, Principle of Brillouin scattering: a small portion of the incident laser light is scattered by spontaneous thermal-induced sound waves (phonons) and gives rise to the Brillouin spectrum. **b**, LSBM set-up: the light is delivered and collected by two objectives mounted in an inverted 90° configuration. The sample is placed in a chamber (attached to a three-axis translational stage for scanning) that is separated from the surrounding immersion liquid by a thin fluorinated ethylene propylene (FEP) foil (inset). The light collected from the sample is redirected to the spectrometer (dashed box), which consists of a combination of spherical and cylindrical lenses to generate the correct beam profile as input for the virtually imaged phased array (VIPA) that disperses the light on the camera. A pure [87]Rb cell rejects the elastically scattered (Rayleigh) light. **c,d**, Scattering geometry for the orthogonal configuration, that is, the O-LSBM (**c**) and the epi-line configuration, that is,

the E-LSBM (**d**). In the orthogonal geometry (**c**) the illumination direction ($k_i$) is perpendicular to the detection ($k_s$), consequently the phonon direction (q) is at 45° to both of them. In the E-LSBM (**d**) only a single objective is used and the illumination direction is parallel to both the detection and the phonon direction. The convention for the $x, y, z$ axes is the same in all of the figures. **e,f**, Characterization of the LSBM in terms of spatial resolution (**e**) (measured by Rayleigh-scattered light from 0.3 μm beads, more details in the Methods section) and spectral precision (**f**), determined by calculating the standard deviation of multiple measurements of Brillouin shift in water in typical experimental conditions (Methods) for both the O- and E-LSBMs. The solid lines in **e** represent Gaussian fits from which the FWHM (full width at half maximum) is extracted. For details concerning the SPIM modality integration, see Extended Data Fig. 1.

## High-resolution and time-lapse Brillouin imaging in ascidians

To further explore the potential of the LSBM to collect high-resolution 3D mechanical data from developing organisms, we imaged the ascidian *Phallusia mammillata*, a simple chordate with highly conserved stereotypic cell lineages and cellular organization (Fig. 3). Here, we used the orthogonal-line configuration of the microscope (O-LSBM) to collect high-resolution, subcellular elasticity information in the early embryo, over an FOV of 165 × 186 × 172 μm (z increment of 2.5 μm) and within ~17 min, shorter than the average time between developmental stages of this animal at 18 °C (Fig. 3a,b). We observed a perinuclearly localized, high Brillouin signal surrounding a large fraction of nuclei in 16-cell-stage cells. (Fig. 3c,d and Supplementary Fig. 1). This highlights the subcellular resolution of our LSBM, although further work is required to pinpoint the exact molecular origin of this signal. We also applied the epi-line configuration (E-LSBM) to a later developmental stage (late tailbud I), to study tissue-level differences of 3D mechanical properties (Fig. 3e–h). Here, with fluorescent SPIM imaging of the lipophilic dye FM4-64 as a membrane marker, we were able to distinguish between the epidermis, central nervous system, and mesoderm and endoderm cells and observed differences in Brillouin shift between them (Fig. 3h), which may suggest that the different nature and organization between the epithelial (epidermis and central nervous system) and mesenchymal cell types (endoderm and mesoderm) results in differential tissue mechanics. These experiments also show that LSBM imaging can be used over long periods

(here, 14 h; Supplementary Videos 3 and 4), thus enabling tracking changes in mechanical properties driven by cellular differentiation over time. No photodamage or phototoxicity were observed at ~18 mW of average laser power, and viability assays showed that all imaged embryos (n = 6) had normal development (as assessed by morphology and behavior).

## Non-invasive Brillouin time-lapse imaging during preimplantation mouse embryo development

Finally, to test the low phototoxicity of the O-LSBM approach in another organism, we imaged the developing mouse embryo, from the 8-cell stage (embryonic day (E)2.75) to the late blastocyst stage (E4.5), covering a 46 hour timespan (Extended Data Fig. 8 and Supplementary Videos 5,6). We acquired 3D Brillouin time-lapse volumes over an ~111 × 131 × 121 μm FOV (z increment of 1.5 μm) within ~11–17 min and at 75–90 min time intervals as well as simultaneous SPIM data that enabled cell tracking via the fluorescently labeled nucleus (H2B-mCherry) and membrane (Myr-Palm-iRFP). Despite the embryos' high photosensitivity, no photodamage or phototoxicity were observed at less than ~15 mW of average laser power as confirmed by the morphology, dynamics, cell number and cell fate of the imaged embryos, which resembled those of the control embryos (Supplementary Videos 5,6 and Extended Data Fig. 8c,d). This represents a substantial improvement over standard confocal Brillouin microscopy with 532 nm illumination, in which embryo death was observed following a few 2D image acquisitions,

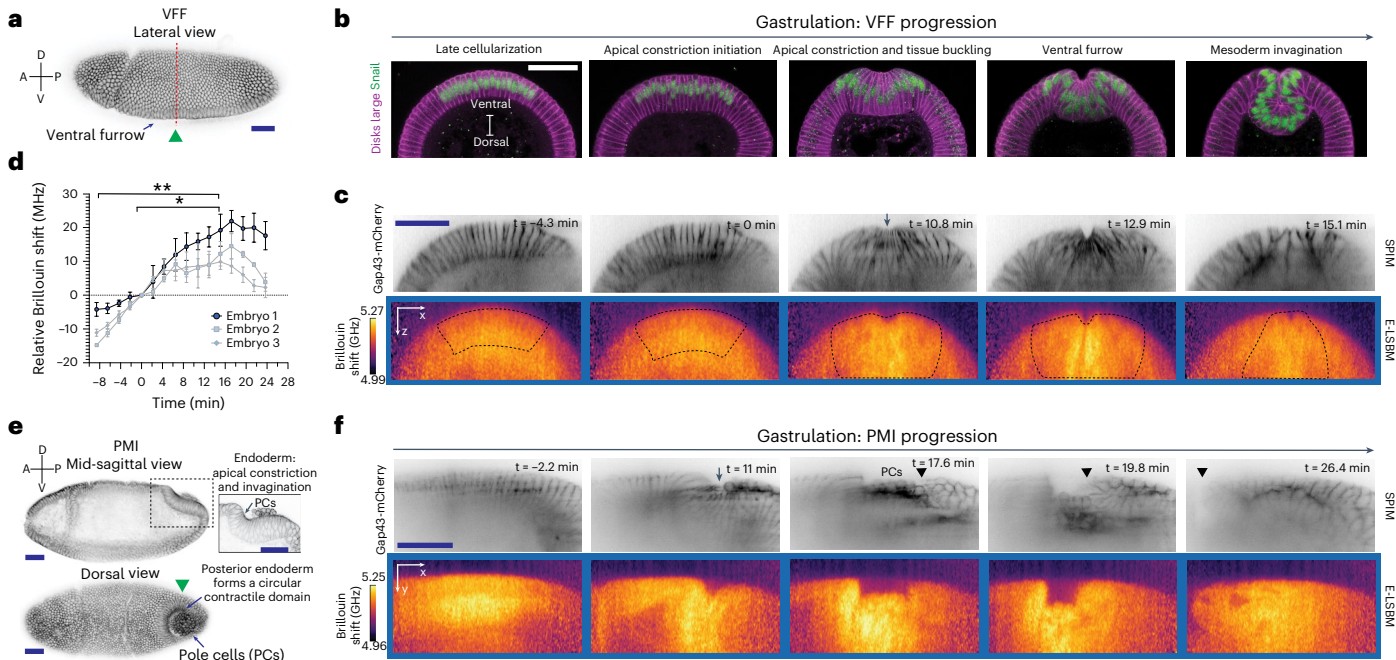

**Fig. 2 | Cells undergoing tissue-folding events during *Drosophila* gastrulation show high Brillouin shifts. a**, Surface view of a fixed *Drosophila* embryo undergoing VFF (arrow). The embryo expresses the membrane marker Gap43-mCherry to visualize cell outlines. The green arrowhead indicates the direction of Brillouin illumination. The red dashed line indicates the approximate location of the physical cross-sections in **b** and **c**. **b**, Stages of VFF progression shown in cross-sections of fixed embryos, stained with a Snail antibody (green) to label the nuclei of cells engaged in VFF and a disks large antibody (magenta) to label cell outlines. **c**, Top: SPIM images of an embryo expressing Gap43-mCherry to mark cell membranes; bottom: median-projected Brillouin shift maps from the same embryo at the same timepoints and positions (representative embryo of *n* = 3 in total). The arrow indicates the formation of the ventral furrow. The dashed box encloses the potential mesoderm region in the Brillouin shift maps used for quantification in **d**. **d**, Quantification of Brillouin shift averaged over 18 cells engaged in apical constriction and invagination during VFF progression, as inferred from SPIM slices (dashed box in **c**). Changes in Brillouin shift are shown relative to the Brillouin shift at timepoint 0 min, which corresponds to the initiation of apical constriction. Data points are average Brillouin shift in the contractile domain, error bars denote s.d. over different slices of each analyzed embryo. Statistical significance was evaluated using a paired one-way ANOVA test (F = 44.29, *P* = 0.066), followed by a post-hoc multiple comparison test (false discovery rate corrected, α = 0.05, q-value = 0.1); *\**P* = 0.039, *\*\**P* = 0.0054 (*n* = 3 embryos from independent experiments). **e**, PMI in embryos at a similar stage as in **a**. Top: mid-sagittal section; inset: higher magnification of the posterior midgut region showing apical constriction and invagination of the posterior midgut primordium; bottom: surface dorsal view of a *Drosophila* embryo showing the formation of the circular contractile domain of the posterior midgut that encloses the pole cells (arrow). **f**, Representative SPIM images (top) and median-projected Brillouin shift maps (bottom) at five timepoints during invagination (out of *n* = 3 in total). The arrow indicates the formation of the circular contractile domain, comparable to the PMI progression in **e** (inset). The arrowhead indicates the displacement of the posterior end during PMI. The initiation of apical constriction was set as the 0 min timepoint. Images in Gap43-mCherry and Brillouin panels are median projections of three slices of the re-sliced region of interest (see Methods). A, anterior; D, dorsal; P, posterior; V, ventral. Scale bars, 50 μm.

---

as indicated by a decrease in Brillouin shift and by the rounding and decompaction of the embryos within 6 hours (Extended Data Fig. 9).

## Discussion

Here, we have shown that entire multicellular organisms can be successfully imaged with Brillouin microscopy at high spatial resolution in three dimensions and over extended time periods without apparent photodamage. Compared with previous implementations[7,9,13,14,18], our LSBM demonstration represents a more than 20-fold improvement in terms of volume imaging speed and lowers the illumination energy per pixel by at least 10-fold, all while achieving similar measurement precision (Fig. 1f and Extended Data Figs. 4d,10). We note that a square root dependence of precision as a function of energy (Extended Data Figs. 4d and 10c,d) is consistent with shot noise-limited conditions. Although the effective pixel time is <1 ms, we note that the fastest dynamics that the LSBM can probe are still limited by the camera exposure time of ~100 ms.

Due to these advances, the LSBM can detect changes in mechanical properties at the cell and tissue scale that occur, for example, during early development in a broad range of organisms. Specifically, we show that a higher Brillouin shift is a common feature of tissue folding, independent of the geometry of the contractile domain (Fig. 2c,d,f). We were also able to visualize subcellular structures with different mechanical properties (Fig. 3a–d), as well as differential tissue mechanics deep inside an embryo (Fig. 3e–h). Achieving this in three dimensions without the need for tissue sectioning (and its concomitant artifacts) or the invasive injection of micro-particles or micro-droplets represents unprecedented capabilities for mechanobiology research.

The fact that our LSBM system can easily switch between two scattering geometries means that the performance can be optimized for a given application. Although the O-LSBM provides the lowest photodamage burden and best axial resolution, optical sample accessibility from two sides must be ensured, and optical refraction effects between the sample and medium can affect image quality. The latter could in principle be mitigated by refractive index matching or adaptive optics methods[29]. In contrast, the E-LSBM geometry is inherently insensitive to these effects, and is thus better suited for more scattering or heterogeneous samples, but has slightly lower axial resolution and a higher illumination light burden for large volumes. We note that although the spatial resolution of our LSBM is substantially higher than previous LSBM implementations[20], it is lower than in confocal Brillouin microscopy[9,14]. In the future, the symmetric design of the microscope

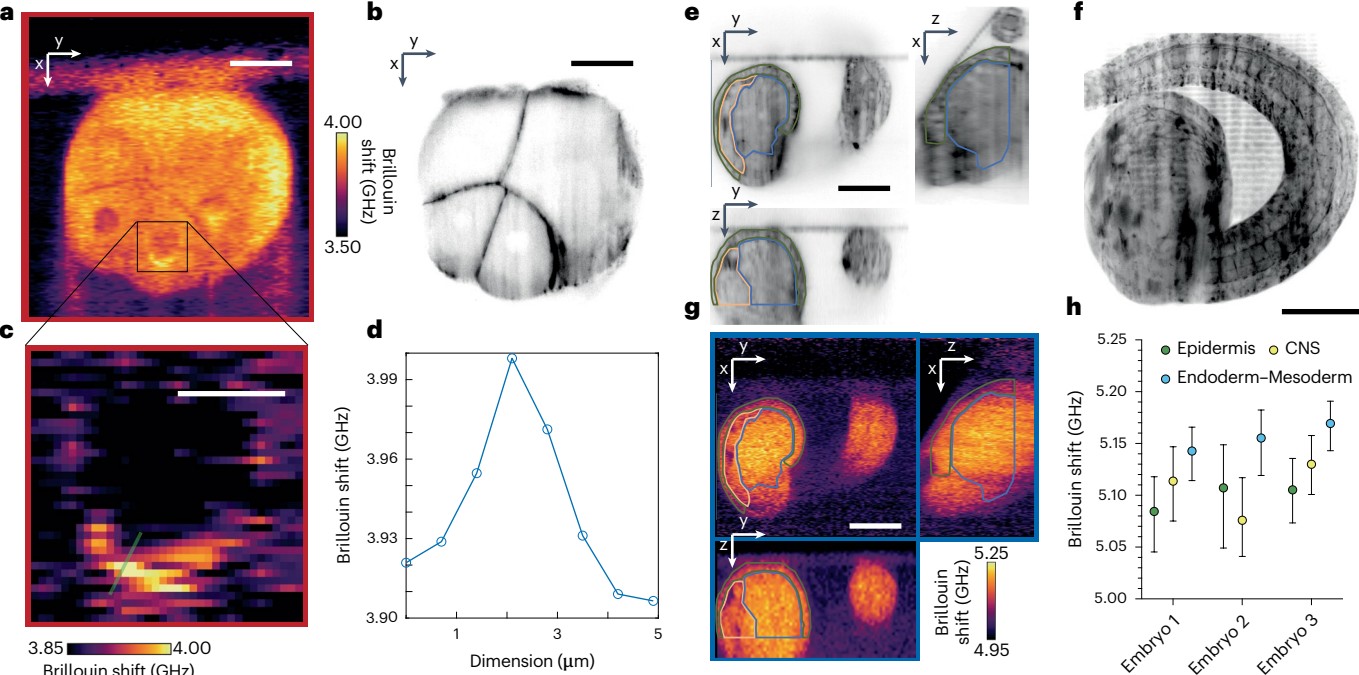

**Fig. 3 | LSBM imaging of *Phallusia mammillata* shows 3D mechanical differences on a subcellular scale and between tissue types over large volumes. a,b**, Representative O-LSBM image (**a**) and corresponding SPIM image (**b**) (membranes labeled with lipophilic dye FM4-64) of a *Phallusia mammillata* embryo at the 16-cell stage (out of *n* = 2 in total). **c**, Zoom-in of the gray square in **a** showing a high Brillouin shift in the vicinity of the nucleus. **d**, Line profile along the line shown in **c. e–h**, Imaging of a *Phallusia* embryo at the tailbud stage (out of *n* = 3 in total). **e,f**, Orthogonal views (**e**) and maximum intensity projection (**f**) from the SPIM volume (192 μm × 190 μm × 111 μm) of membranes labeled with lipophilic dye FM4-64 of a *Phallusia* embryo at the tailbud stage (late tailbud I). **g**, Corresponding orthogonal views, acquired with the E-LSBM, of the volume

in **e. h**, Median Brillouin shift averaged volumetrically over three distinct tissue regions (epidermis, central nervous system (CNS) and endoderm–mesoderm) in *n* = 3 embryos, manually segmented in 3D according to SPIM data. Embryo 3 is shown in **g**. There was no significant difference in Brillouin shift between the segmented embryonic domains (*P* = 0.1944, one-way ANOVA, paired, non-parametric test (Friedman test)). Each dot represents the median, and the lines represent the interquartile range of segmented domains in each analyzed embryo (*n* = 3 embryos from independent experiments). The green, yellow and light-blue outlines in **g** and **h** show the segmented regions for epidermis, CNS and endoderm–mesoderm, respectively. Scale bars: **a,b**, 30 μm; **c**, 10 μm; **e–g**, 50 μm.

described here can be used for rapid, dual-view imaging by optically switching the illumination and detection paths, an approach that could mitigate shadowing effects and improve resolution through data fusion[30].

To conclude, we expect that the low phototoxicity and significantly improved acquisition speed of the LSBM will open up exciting possibilities for the field of mechanobiology. In particular, we anticipate applications in the imaging of mechanical changes during development together with, for example, cytoskeletal or cell fate-specific fluorescence reporters, or by correlating 3D mechanical with molecular, genetic or ultra-structural datasets[31]. This would be able to shed light on the complex interplay between genetics, biochemical signaling and the role of biomechanics in animal development.

## Online content

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

## Methods

### Line-scan Brillouin imaging set-up

A detailed schematic diagram of our LSBM is shown in Extended Data Fig. 1. The laser is an amplified tunable diode laser (TA pro, Toptica) locked to the $D_2$, F = 2 absorption line of $^{87}$Rb (780.24 nm). To clean the laser's spectrum to remove amplified spontaneous emission (ASE) noise, we designed a custom-built narrowband double-pass optical filter[32], consisting of a polarizing beam splitter followed by a tunable Fabry–Pérot cavity (free spectral range (FSR) 15 GHz, Finesse ~60, Light-Machinery), a holographic diffraction grating (NoiseBlock, Coherent), a quarter waveplate and a mirror. A beam sampler (BSF10-B, Thorlabs) sends ~5% of the light to a photodiode, the signal from which is fed into Arduino Uno SMD R2 and PmodDA3 digital to analog converter boards; a self-written Arduino 1.8.13 code maximizes the transmitted intensity to correct for long-term drift of the cavity (Extended Data Fig. 2a). The filter enables a total suppression of the integrated ASE noise down to approximately −90dB, more than 40 dB below the original ASE level of the laser (Extended Data Fig. 2c). After the filter, the light is coupled into a polarization-maintaining fiber and delivered to the main microscope. Here, a collimator (LPC-OT-780-5/125-P-3-16AC-40-3A, OZ Optics) produces a collimated beam with a diameter of 2.77 mm ($1/e^2$, theoretical), which corresponds to a focused beam of 1.79 μm ($1/e^2$, theoretical) after the objective. The beam then passes through an electrically tunable lens (EL-10-30, Optotune) coupled with a negative offset lens (LC1258-B, Thorlabs). This enables axial scanning of the focus position after the objective (Extended Data Fig. 6). After the tunable lens, a flip mirror enables switching between the epi-line and orthogonal-line illumination: the separate optical path for the epi-line illumination consists of a 40 mm lens (LA1422-B, Thorlabs) that collimates the light out of a polarization-maintaining fiber (P3-780PM-FC-2, Thorlabs) to provide a beam of diameter 8.7 mm ($1/e^2$, theoretical) (filling the back focal aperture of the objective, which is 8 mm); a 200 mm cylindrical lens (LJ1653RM-B, Thorlabs) generates the focused line in the focal plane of the objective with an extension of ~185 μm ($1/e^2$) (Extended Data Fig. 6). A half waveplate (WPH05M-780, Thorlabs) is used to rotate the linear polarization of the illumination light to match the required direction for the two geometries. Two mirrors, conjugated to the back and the front focal plane of the objective (by means of two 200 m lenses: cat. nos. 33-362 and 49-364, Edmund), respectively, enable fine position and angle adjustment of the beam inside the sample to aid in the alignment between the illumination and detection objectives.

For the fluorescence SPIM modality, we use a laser box providing several laser lines for fluorescence excitation (405 nm, 488 nm, 560 nm, 640 nm). The fluorescence excitation is introduced into the set-up by a single mode fiber (P1-460Y-FC-1, Thorlabs), the output of which is collimated by an achromatic lens (f = 4 mm), to produce an ~0.8 mm ($1/e^2$, theoretical) beam. A fiber polarization controller (FPC030, Thorlabs) is used to adjust the polarization and a polarizer rejects potential residual non-linear polarization. The excitation light is then coupled into the same optical path as the Brillouin illumination by means of a dichroic mirror (FF765-Di01, Semrock). Similarly to the Brillouin illumination path, two lenses (67-159, Edmund and 49-364, Edmund, shared with the Brillouin path) and mirrors (not shown in Extended Data Fig. 1) enable beam adjustment. An 18 mm cylindrical lens (68-041, Edmund) is used to generate the light sheet. The corresponding thickness and lateral extent of the light sheet inside the sample are ~4 μm and ~200 μm ($1/e^2$, theoretical), respectively.

The detection and illumination objectives are identical (Nikon, ×40, 0.8 numerical aperture, MRD07420, water immersion) and oriented in an inverted V configuration to facilitate the imaging of small (~200 μm) samples in physiological conditions[21] (Fig. 1a and Extended Data Fig. 1). The sample is placed in a dedicated, custom-built chamber that utilizes a 12.7-μm-thick fluorinated ethylene propylene (FEP) foil (RD-FEP050A-610, Lohmann) to separate the immersion water of the objectives from the sample. The sample's environment (medium) is temperature controlled via a custom-built heating element and PID (proportional–integral–derivative) controller, and $CO_2$ is added via a custom-built gas-mixer (Extended Data Fig. 1c,d). For 3D image acquisition, the chamber containing the sample is scanned by three single-axis translational stages (SLC-2430, SmarAct).

For the epi-line Brillouin geometry (E-LSBM), a polarizing beam splitter (BPB-12.7SF2-R400-800, Lambda), mounted on a magnetic base for easy switching between the two modalities, followed by a quarter waveplate (WPRM-25.4-12.7CQ-M-4-780, Lambda), reflects the light towards one of the two objectives; the Brillouin signal is collected by the same objective and transmitted by the polarizing beam splitter towards the spectrometer (the polarization is rotated by 90° due to the double pass at the quarter waveplate). A polarizer is added before the other objective (used for SPIM excitation), which is required to block the backwards-scattered un-polarized fluorescence light that would otherwise be reflected by the polarizing beam splitter, and generate a bright line at the center on the FOV of the SPIM camera.

The fluorescence and Brillouin signals are split via a dichroic mirror (FF765-Di01, Semrock), reflecting the fluorescence emission. A tube lens (MXA20696, Nikon) focuses the fluorescence image on the SPIM camera (Zyla sCMOS, Andor); a motorized filter wheel (96A361, Ludl) enables selection of the proper emission filter. The Brillouin signal is instead transmitted by the dichroic mirror and sent to the spectrometer (Extended Data Fig. 3). Here, first two spherical lenses (LA1256-B and LA1979-B-ML, Thorlabs) relay the back focal aperture of the detection objective and a slit (VA100/M, Thorlabs), placed in the focal plane of the first lens, ensures confocality in the direction perpendicular to the illumination line. After the relay, inelastically scattered (Rayleigh) light is filtered by a 150-mm-long gas cell filled with pure $^{87}$Rb (SC-RB87-(25×150-Q)-AR, Photonics Technologies). The cell is wrapped into heating foil (HT10K, Thorlabs) and a self-built electronic board (based on an Arduino Mega) controls the temperature of the cell via PID. After the rubidium gas cell, a combination of an achromatic lens (cat. no. 49-794, Edmund), a cylindrical lens (cat. no. 36-231, Edmund) and a spherical lens (cat. no. 69-513, Edmund) relay and focus the light onto the entrance slit of the virtually imaged phased array (VIPA; OP-6721-6743-9, LightMachinery). Between the second spherical lens and the VIPA there is an additional 75-mm-long $^{87}$Rb cell (SC-RB87-(25×75-Q)-AR, Photonics Technologies).

After the VIPA two cylindrical lenses (LJ1267L1-B and LJ1629L2-B, Thorlabs) focus the light on the EMCCD (electron-multiplying charge-coupled device) camera (iXon DU-897U-CS0-BV, Andor), preceded by a bandpass filter (FBH780-10, Thorlabs) to block ambient light. These lenses are chosen to ensure proper spatial and spectral sampling (0.7 μm px$^{-1}$ and ~0.25 to ~0.5 GHz px$^{-1}$). The optical layout and performance are shown in Extended Data Figs. 3–5. To synchronize the electrically tunable lens with the EMCCD acquisition, the fire output of the EMCCD (high when the camera is exposing) is used to trigger a data acquisition device (PCI-6221, National Instruments) that generates the voltage ramp driving the electrically tunable lens. The entire data acquisition is controlled by a self-written LabView2018 program and the image reconstruction is performed with a Matlab2018 script, which provides an interface to the GPU-fitting library (see the section Brillouin line-scan spectral analysis).

### LSBM system characterization and image acquisition

To quantify the optical resolution of the O-LSBM and E-LSBM modalities, we embedded 0.3 μm fluorescent beads (TetraSpeck, Thermo Fischer) in 1% agarose and recorded the Brillouin spectrum while scanning the sample along the $y$ direction with a step size of 0.1 μm for O-LSBM and E-LSBM, and along the $z$ direction with a step size of 0.1 μm for O-LSBM and 0.25 μm for E-LSBM. By assigning the amplitude of the Rayleigh peak to the corresponding spatial position, Extended Data Fig. 4a and Extended Data Fig. 4b were generated for the O-LSBM and E-LSBM modalities, respectively. The plots in Fig. 1e were generated

by summing the intensity in Extended Data Fig. 4a,b along the direction perpendicular to the one shown. To quantify the spatial pixel size we moved the bead along $x$ and performed a linear fit (Extended Data Fig. 4c). To quantify the spectral precision we collected 50 spectrometer images (lines) from water in typical imaging conditions (100 ms integration time, less than ~18 mW), selected the spectra from the center of the FOV (where the intensity is approximately constant), determined the Brillouin shift after applying the reconstruction pipeline, and calculated the histogram (Fig. 1f) and the standard deviation, To measure the spectrometer's spectral resolution, we collected the Brillouin signal from scattering intralipid solution, remapped and summed the spectra (see the section Brillouin line-scan spectral analysis) and performed a Lorentzian fit on the Rayleigh signal. The signal-to-noise ratio in a biological sample (Extended Data Fig. 10) was calculated as the ratio of the amplitude of the Stokes peak divided by the standard deviation of the residuals after fitting the spectra with the appropriate function (that is, the difference between the raw data and the fit). The ASE was measured with the set-up shown in Extended Data Fig. 2b, which consists of the addition of a diffraction grating to the Brillouin spectrometer to ensure an extended FSR while retaining high spectral resolution. The rubidium cells suppress the laser light, thus allowing for the very high dynamic range (>90 dB) needed for these measurements.

## Brillouin line-scan spectral analysis

Each spectral line (corresponding to the light scattered from a point along the illumination line) is analyzed separately. The processing of this spectral data is then done in three main steps. First, the spectrum is remapped into a linear frequency space; second, the signal from at least three different orders is summed to increase the effective signal-to-noise ratio; and last, the Stokes and anti-Stokes peaks are fitted separately. To speed up the entire processing via parallelization, we decided to implement this analysis pipeline with custom GPU code written in CUDA. The linear remapping (pixel-to-frequency conversion) depends only on the position of the Rayleigh peak and the FSR of the VIPA (15.15 GHz). Here, the FSR was measured from the calibration of the spectrometer with rubidium $D_2$ hyperfine lines of known frequency (Extended Data Fig. 5a,b). For the remapping we use spline interpolation whenever the target frequency sampling does not match the frequencies of individual pixels. Each line of the image can be processed independently, enabling this operation to be executed in parallel on the GPU. After this pre-processing step, the individual spectra are fitted over a user-selected range using GPUfit[33]. Lorentzian, quadratic and broadened Brillouin line shape[25] functions were custom added into this library. In addition to the main Brillouin peak position and linewidth, other fitting parameters, such as amplitude, signal-to-noise ratio, goodness of fit, and number of iterations can be retrieved. The reported Brillouin shift is the average between the frequency shift of the Stokes and anti-Stokes peaks.

To benchmark our GPU pipeline, we created a synthetic Brillouin spectral signal, which we then processed with either the standard, CPU-based Matlab script or our custom GPU code. On our personal computer (Intel Core i9-11900K, 64 GB RAM and Nvidia GeForce GTX 1050 Ti), the GPU processing took ~10 ms regardless of the dimension of the image, while the purely Matlab-based pipeline took more than ~1 s, and scaled linearly with the image size (Extended Data Fig. 7a). Both processing methods give similar results, with negligible (~1 MHz) difference between the fitted peak frequencies and the real value (Extended Data Fig. 7b).

## Drosophila embryo imaging

**Live imaging.** *Drosophila* live imaging was performed with embryos expressing Gap43::mCherry to label cell membranes and EB1-GFP to mark microtubules plus ends[34]. Embryos for Brillouin imaging were prepared as follows: 1 h synchronized egg collections were performed at 25 °C and embryos were allowed to develop for a further 1 h at 25 °C.

Next, halocarbon oil (27S, Sigma) was added on the agar plate carrying the collected embryos to enable accurate embryonic staging through visual inspection of morphological features under a binocular microscope (Zeiss). Embryos in stage 5a-b[35] were hand-selected, dechorionated in sodium hypochlorite (standard bleach, 50% in $H_2O$) for 90 s, and washed using PBS 1X. Finally, embryos were glued (with heptane glue) to the outer surface of the FEP film in the desired orientations to enable the collection of data from different angles. Image acquisition was initiated 5 min after the cellularizing front of the blastoderm cells passed the basal side of the nuclei. The temperature of the imaging chamber was set to 22 °C.

***Drosophila* Snail antibody generation.** A construct encoding an amino-terminal glutathione S-transferase (GST)-tagged portion of the *Drosophila* Snail protein (encompassing amino acids 1–200 of Uniprot ID P08044) was expressed in *Escherichia coli* and purified from lysates of bacteria expressing the Snail–GST fusion protein using beads with immobilized glutathione. The fusion protein was eluted from beads using reduced glutathione and used for immunizing a rabbit. The immunization and collection of pre-bleeds and test bleeds was performed by Cocalico Biologicals. The antiserum was tested for specificity and titer using immunostaining, immunoprecipitation and western blot analysis.

**Immunohistochemistry.** *Drosophila* whole mount embryos were fixed and stained as previously described[36]. In Fig. 2a,e, Gap43-mCherry transgenic embryos were stained using an mCherry antiserum (rabbit, 1:1,000, Abcam ab167453). In Fig. 2b, w[1118] embryos were co-stained with the anti-Snail antibody described above (Rabbit, 1:500) and an anti-disks large antibody to label the lateral membrane of cells (mouse, 1:50, 4F3, DSHB). Physical cross-sections across half of the embryonic length were produced on these stained embryos as previously described[37].

**Image processing and automated image analysis.** To enable SPIM fluorescence image-guided analysis, the Brillouin time-lapse images were first re-scaled (bilinear interpolation) in the $x$ and $y$ directions using a factor of 2.5138 to match them to the dimensions of the fluorescence images. The fluorescence time-lapse was manually aligned with the Brillouin time-lapse in the $y$ direction using half the diameter of the fluorescence image (450 pixels), and in the $x$ direction using the co-localization of the furrow as the landmark (either the VFF or PMI furrow). The aligned fluorescent time-lapse was cropped using the Brillouin dimensions to stack the two time-lapses and produce an overlay image. For VFF image processing and quantification, for each analyzed embryo the overlaid time-lapse was re-sliced and three median projections of two slices each were used as embryonic replicates. To conduct the automated image analysis, we manually generated masks that included an 18-cell domain containing the mesoderm (Fig. 2b, late cellularization, Snail-positive cells). The region enclosed by the mask was determined by taking nine cells in each direction (left and right) from the center of the furrow. These masks were loaded by a Matlab script that filtered the Brillouin shift maps in each corresponding timepoint for each replicate and for each embryo. The average Brillouin shift in the filtered, potential mesodermal, domain and the corresponding standard deviation were calculated per embryo and plotted against time, taking time = 0 as the first sign of apical constriction. Image quantification was performed for three independent embryos undergoing VFF.

**Statistical analysis.** Prior to selection of the statistical test, we tested for normality using a Shapiro–Wilk test and Q-Q plot; if normality was not rejected, then parametric tests were used. To assess the statistical significance of the differences of the Brillouin shift in Fig. 2d, the average Brillouin shift in the segmented domain for the three independent

embryos at timepoints −8.6 min, 0 min and 15.1 min was analyzed using a paired one-way ANOVA, followed by a post-hoc multiple comparison test (corrected for false discovery rate, using the two-stage step-up method of Benjamini, Krieger and Yekutieli, q-value = 0.1). Sphericity was not assumed (equal variability of differences). Statistical significance was set at α = 0.05. To analyze the statistical significance of the Brillouin shifts in Fig. 3h, we compared the medians between the segmented cell populations using a Friedman test. Statistical analyses were performed using Prism GraphPad v8-9.

### *Phallusia* imaging

**Live imaging.** Adult *Phallusia mammillata* were provided by the Roscoff Marine Biological Station (France) and kept at 17 °C under constant illumination. In vitro fertilized embryos were prepared as described previously[38]. Individual membranes were labeled with and imaged in artificial seawater with 5 µg ml$^{-1}$ lipophilic dye FM4-64 (Invitrogen). The imaging chamber of our LSBM system was kept at 12 °C and 17 °C, respectively, in Fig. 3a,b and Fig. 3e–g.

**Image data analysis.** The registration of the SPIM and Brillouin images for analysis was done similarly to that for the *Drosophila* embryo, that is, the Brillouin image was upscaled (bilinear interpolation) to match the pixel size of the fluorescence image. A rectangular area of the same size as the Brillouin image was manually aligned around the center of the fluorescence image (using the edges of the embryo, visible in both modalities, as a guide) to match the Brillouin image. The boundaries of different tissue types were manually drawn, slice by slice, by looking at the morphology of the cells in the fluorescence image. The segmentation masks were then imported into Matlab and used to generate the plot in Fig. 3h.

### Mouse embryo imaging

**Recovery and culture.** Animal work was performed in the animal facility at the European Molecular Biology Laboratory, with permission from the institutional committee for animal welfare and institutional animal care and use (IACUC), under protocol number 2020-01-06RP. The animal facilities are operated according to international animal welfare rules (Federation for Laboratory Animal Science Associations guidelines and recommendations). (C57BL/6xC3H) F1 mice from 8 weeks of age onwards were used. Embryos were recovered from superovulated female mice mated with male mice. Superovulation was induced by i.p. injection of 5 IU pregnant mare's serum gonadotropin (Intervet Intergonan), followed by i.p. injection of 5 IU human chorionic gonadotropin (Intervet Ovogest 1500) 44–48 h later.

**Imaging.** The sample holder was covered with pre-equilibrated (5 % $CO_2$, 37 °C) G-1 Plus medium (~30 µl) and covered with pre-equilibrated Ovoil (~150 µl). Pockets were cast into the membrane as previously described[39]. Embryos were then mounted into the pockets for imaging.

**Immunofluorescence staining.** Embryos were washed twice with PBS, fixed using 1% PFA and 0.1% Triton X-100 (Sigma, T8787) in PBS for 30 min at room temperature, and then washed again three times with PBS. All these steps were carried out in IBIDI slides coated with 0.5% Agar. After washing, the embryos were transferred to uncoated IBIDI slides containing blocking buffer (PBS with 0.1% Triton X-100, 3% BSA (Sigma, 9647) and 5% normal goat serum) and incubated overnight at 4 °C. After blocking, the embryos were incubated overnight at 4 °C with primary antibodies diluted in blocking buffer. Afterwards, the embryos were washed once with 0.1% BSA and 0.1% Triton X-100 in PBS, incubated for 1 h at room temperature with secondary antibodies and DAPI (Invitrogen, D3751; 1:2,000) diluted in blocking buffer, and washed again with 0.1% BSA and 0.1% Triton X-100 in PBS. Finally, the embryos were mounted individually in 0.5 µl drops of 0.1% BSA in PBS and placed in LabTek chambers covered with oil. Primary antibodies against CDX2

(Biogenex, NC9471689; 1:150), SOX2 (Sigma, SAB3500187; 1:50) were used in this study. For the secondary staining, antibodies targeting mouse immunoglobulin coupled to Alexa Fluor 488 (Life Technologies, A21202), Alexa Fluor 555 (Life Technologies, A31570), and rabbit immunoglobulin-coupled Alexa Fluor 546 (Invitrogen, A10040) were used. Finally, Alexa Fluor 633-coupled phalloidin (Invitrogen, R415; 1:50) was used to stain filamentous actin.

An overview of all imaging parameters used during the LSBM imaging of flies, ascidians and mouse embryos is given in Supplementary Table 1.

### Reporting summary

Further information on research design is available in the Nature Portfolio Reporting Summary linked to this article.

### Data availability

The raw datasets generated and/or analyzed for this work are available at https://doi.org/10.5281/zenodo.7525851. Source data are provided with this paper.

### Code availability

The GPU-accelerated spectral analysis code is open-source under a GPLv3 license and can be accessed at https://github.com/prevedel-lab/brillouin-gpu-acceleration.

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

### Acknowledgements

The authors thank the mechanical workshop and A. G. Ortiz and C. Kieser from the electronic workshop at EMBL Heidelberg for help; T. Hiiragi and L. Hufnagel for support in the early stages of the project; I. Schneider and K. Watanabe for help with mouse imaging experiments; S. Lembo for help with confocal imaging of the stained embryos; and EMBL LAR for mouse husbandry and ascidian support. The authors also thank F. Paul Ukken for the generation of the *Drosophila* Snail antibody and E. Vogelsang for producing physical cross-sections of *Drosophila* embryos; S. De Renzis (EMBL) and N. Bulgakova (University of Sheffield) for providing the sqh-Gap43::mCherry and EB1-GFP transgenic line, respectively; and F. Yang for fruitful discussions. A.D.-M. was supported

by the Deutsche Forschungsgemeinschaft (DFG) research grants DI 2205/2-1 and DI 2205/3-1. The laboratory of M.L. was funded by EMBO and DFG grant LE 546/12-1. R.P. acknowledges support of an ERC Consolidator Grant (no. 864027, Brillouin4Life), and the German Center for Lung Research (DZL). R.P. and A.D.-M. acknowledge funding from the COST Action CA16124 ('BioBrillouin'). This work was supported by the European Molecular Biology Laboratory.

## Author contributions

R.P. conceived the project and designed the imaging system together with C.B. C.B. built the imaging system and wrote the control software. C.B. performed experiments and analyzed data together with C.J.C, J.M.G. and U.-M.F., under the guidance of A.D.-M., M.L and R.P. L.W. designed the opto-mechanics. S.H. wrote the GPU spectral analysis code. J.M.G. provided transgenic fly lines and analyzed data, under the guidance of M.L. M.E. provided and injected transgenic mouse embryos under the guidance of J.E. R.P. led the project and wrote the paper with input from all the authors.

## Funding

## Competing interests

J.E. is scientific co-founder and advisor of Luxendo (part of Bruker), which makes light-sheet-based microscopes commercially available. All other authors have no competing interests.

## Additional information

**Extended data** are available for this paper at https://doi.org/10.1038/s41592-023-01822-1.

**Correspondence and requests for materials** should be addressed to Robert Prevedel.

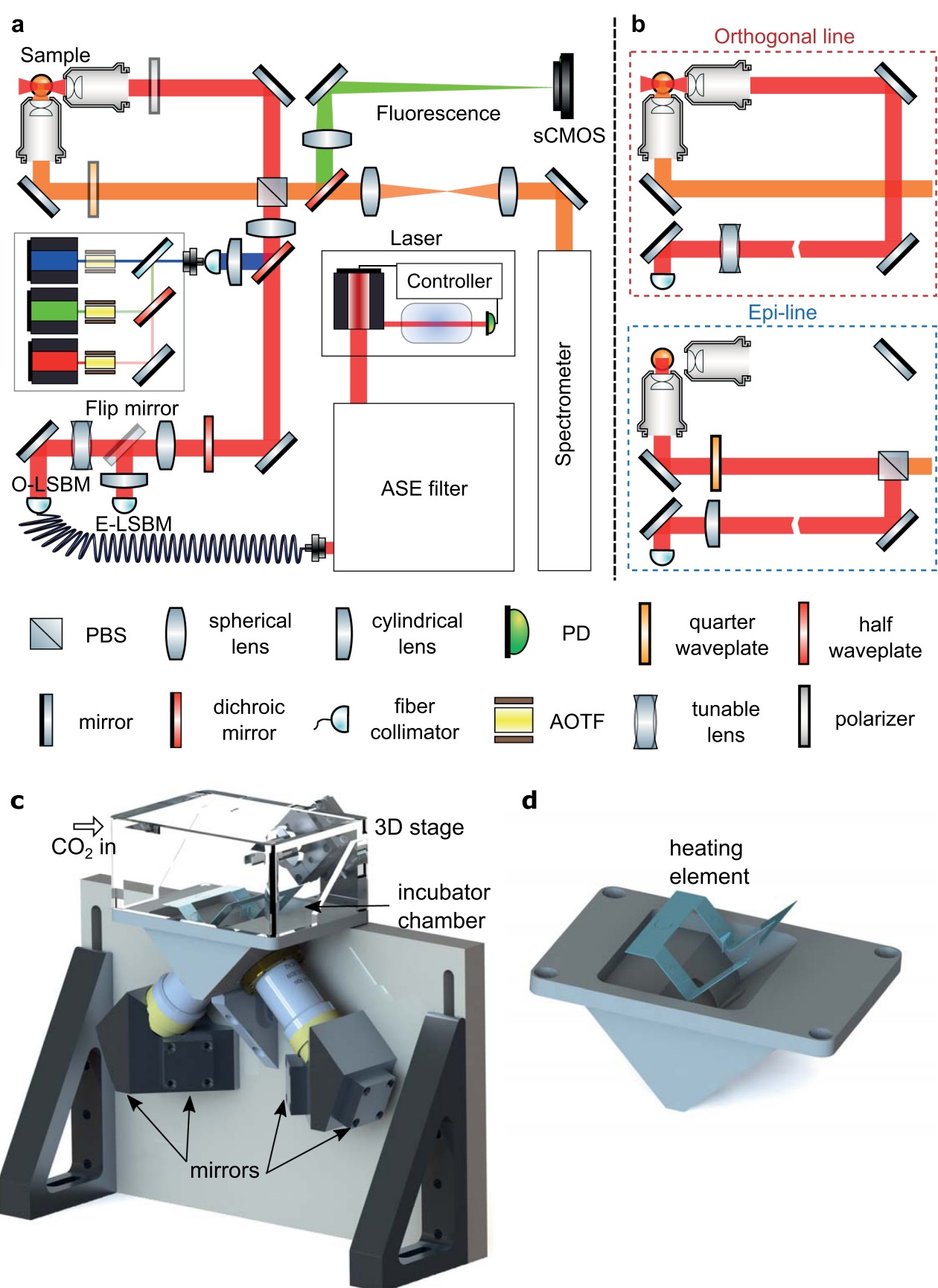

**Extended Data Fig. 1 | See next page for caption.**

**Extended Data Fig. 1 | Optical and mechanical design of the LSBM.**
**(a)** Conceptual schematic of the optical layout including both the fluorescence and Brillouin imaging modalities. The ASE filter and spectrometer are further detailed in Extended Data Figs. 2 and 3, respectively. **(b)** Optical path for the Brillouin illumination and detection in the orthogonal (top) and epi line (bottom) configuration. Note that switching between these configurations can be done by flipping the mirror after the tunable lens, adding/removing the PBS (mounted on a magnetic base to ease this operation) and rotating the half and quarter waveplates appropriately. **(c)** 3D rendering of the mechanical design. The objectives are arranged in an inverted 90 deg V-configuration; light is delivered to (and collected from) the objectives by means of two mirrors that preserve image orientation. **(d)** Enlarged view of the immersion chamber, showing the heating element inset used for temperature control. See Methods for further details.

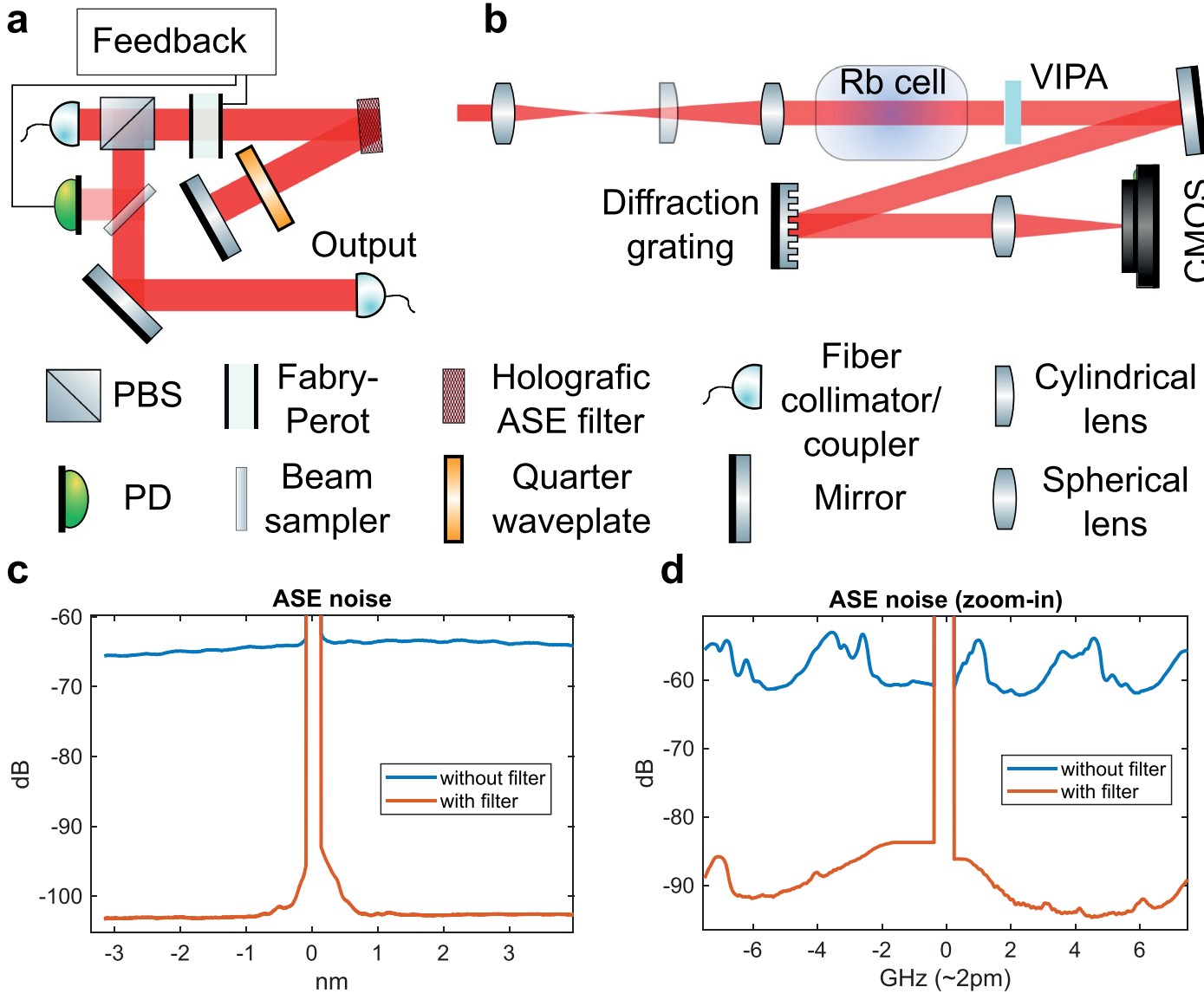

**Extended Data Fig. 2 | Characterization and filtering of Amplified Spontaneous Emission (ASE). (a)** Optical layout of the double-pass ASE filter. To stabilize the Fabry-Perot (FP) based filter over a long time a feedback-loop is realized via a photodiode, custom-written Arduino software and piezo that adjusts the FP mirror spacing. **(b)** Optical set-up used to measure the ASE **(c,d)** Plots of ASE spectrum with and without filter few nm **(c)** and few GHz **(d)** away from the laser line. The filter suppresses the total ASE (integrated over the whole measured range) by more than 40 dB. See Methods for further details.

**a**

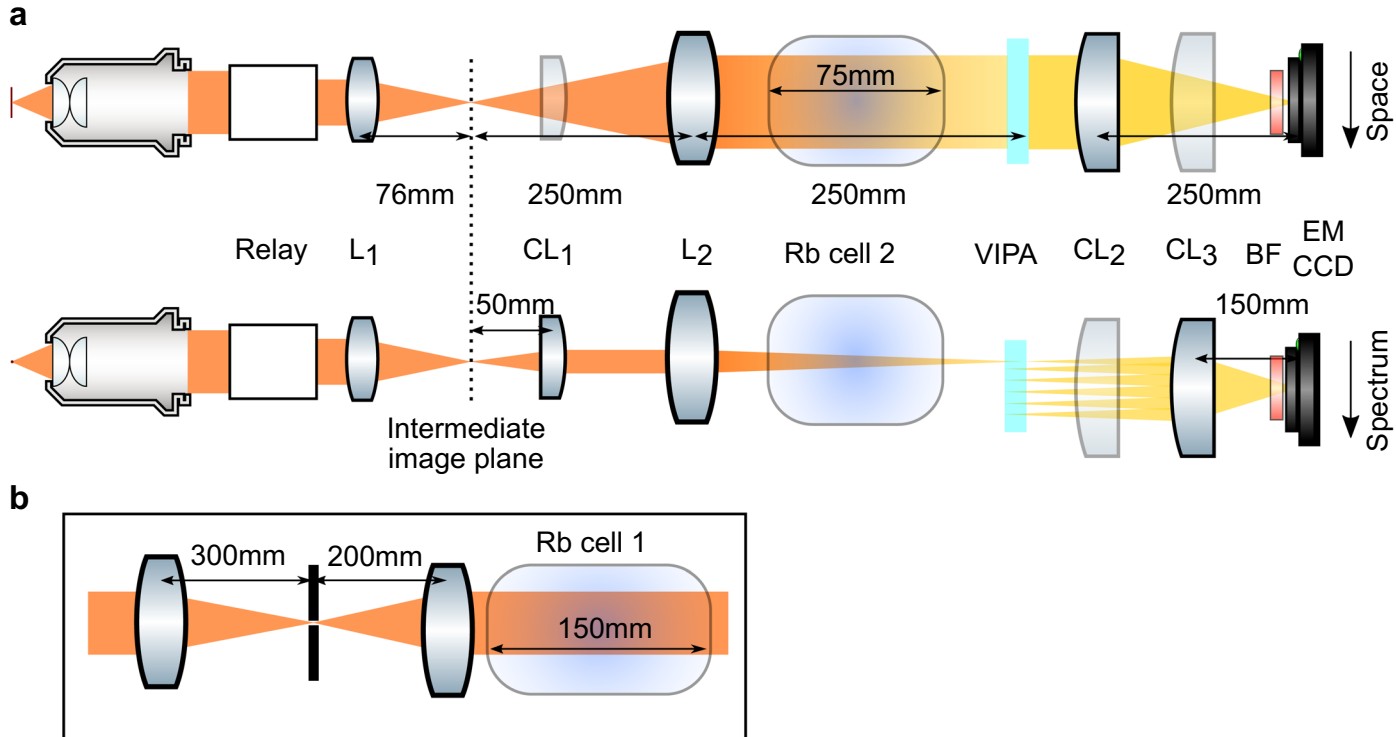

**b**

**Extended Data Fig. 3 | Optical design of the LSBM spectrometer.**
**(a)** Orthogonal views of the optical components in the LSBM spectrometer with relevant distances; top shows the plane determined by the illumination line and the optical axis of the detection objective, where the optics relay to the VIPA and appropriately magnify the back focal aperture of the objective; bottom shows the plane perpendicular to the illumination line, where the optics relay to the VIPA and appropriately magnify the image plane. **(b)** Details of the relay shown in panel **a**.

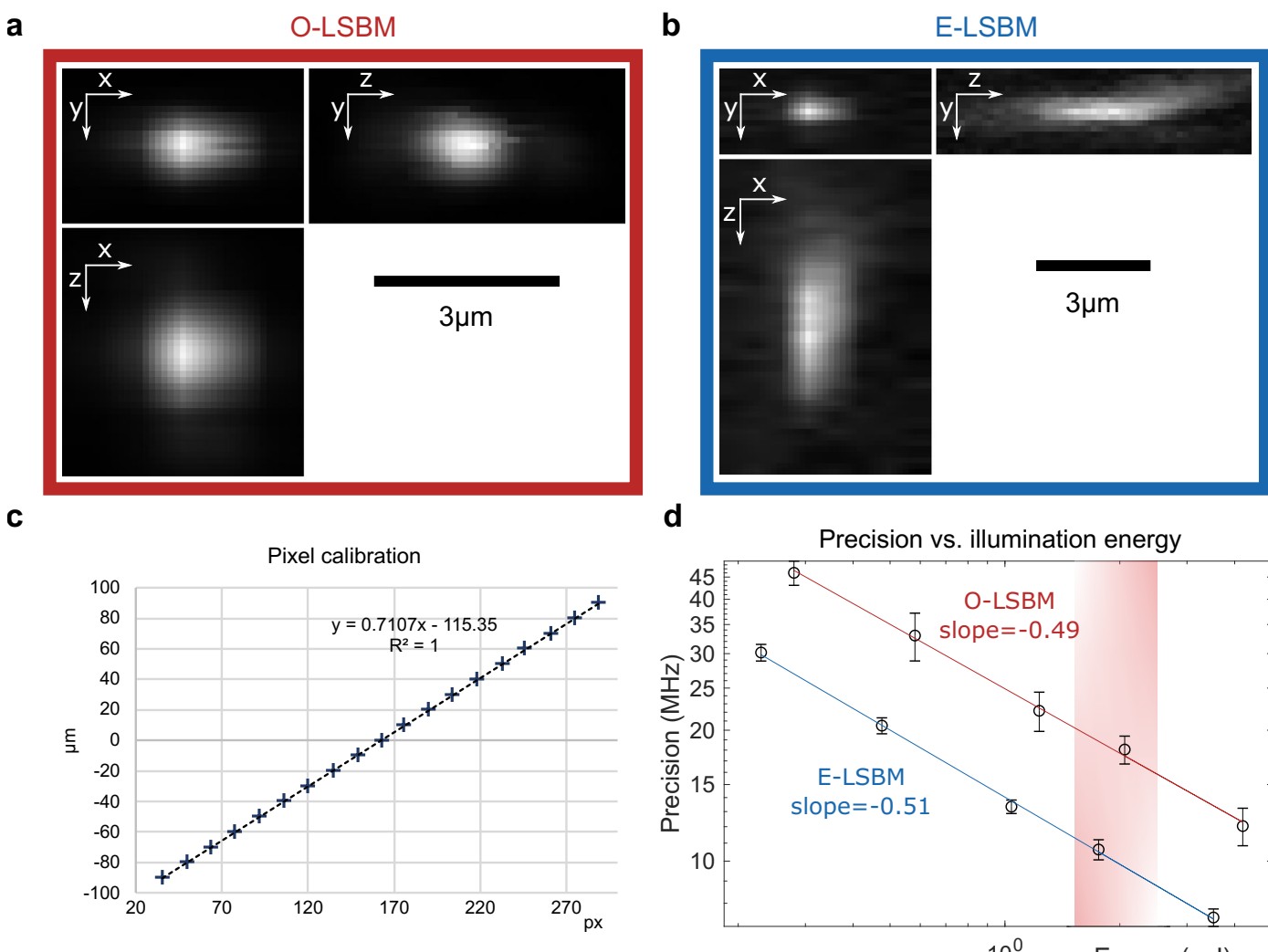

**Extended Data Fig. 4 | Optical characterization and precision measurement of the LSBM spectrometer.** Orthogonal views of a 0.3um bead (Tetraspeck, Thermo Fischer) to characterize the optical resolution in the orthogonal **(a)** and epi line **(b)** geometries. **(c)** Spatial calibration (µm to pixel) acquired by translating a bead, with steps of 10 µm, along the x direction and determining its position on the camera. **(d)** Precision vs. illumination energy for the orthogonal (red) and epi (blue) line geometries. The shaded red region indicates the typical imaging conditions used in the experiments. To generate the precision plot for the O-LSBM (E-LSBM), 441 (441) spectrometer images were acquired from water, each containing more than 200 spectra (one for each spatial position); the data points (circles) represent the precision, calculated as the S.D. of the Brillouin shift extracted from all the spectra selected from 61 (31) at the center of the FOV – that is 26901 (13671) spectra in total; the error bars represent the S.D. of the precision calculated as mentioned before but separately for each point along the FOV– that is the S.D. of 61 (31) values. See Methods for details.

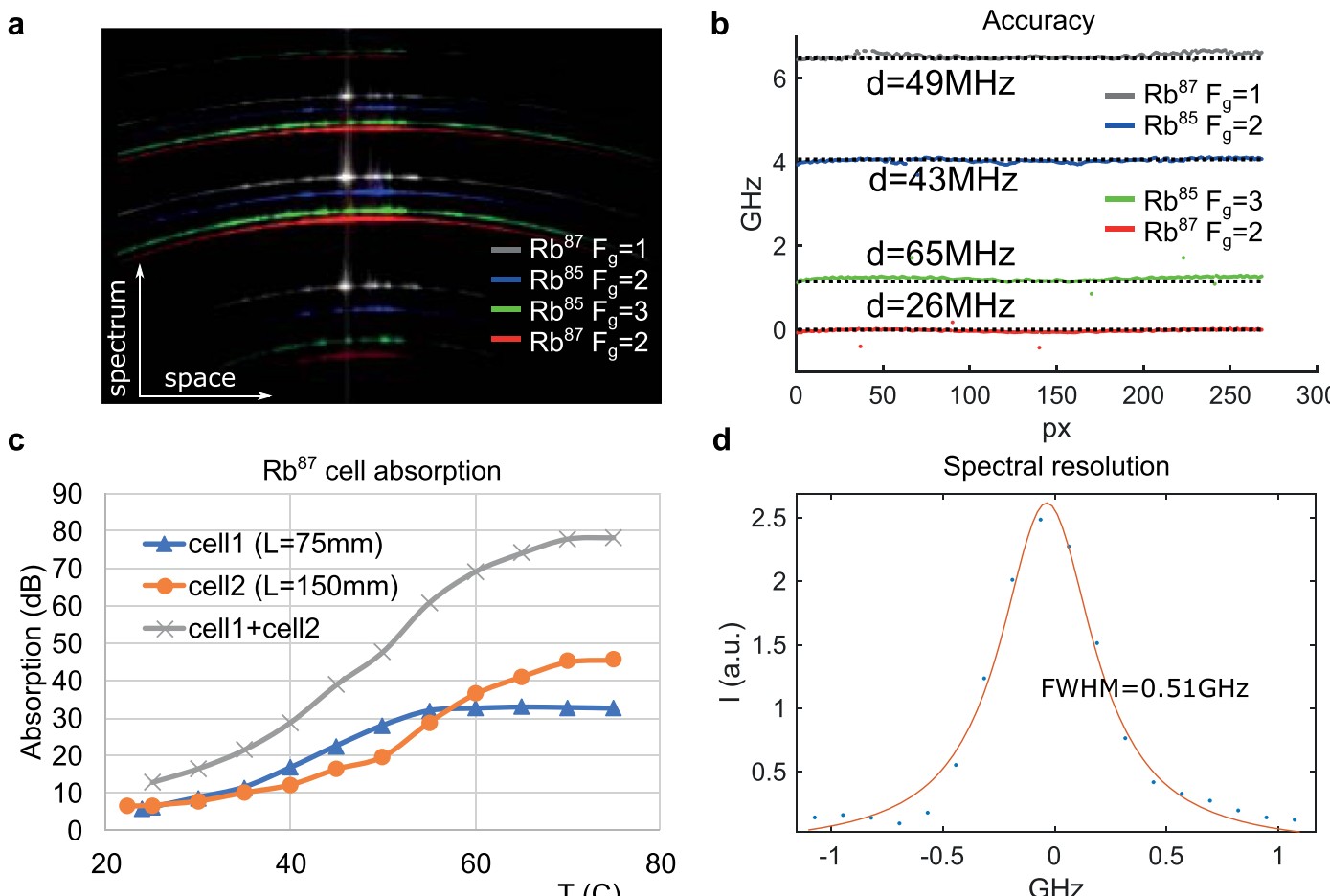

**Extended Data Fig. 5 | Spectral characterization of the LSBM spectrometer.** (a) Acquired, raw spectra of the 4 hyperfine D2 spectral lines of $Rb^{85}$ and $Rb^{87}$ as measured by the LSBM spectrometer. (b) Frequency shift of the Rb lines in panel **a** after applying the reconstruction pipeline (dotted lines represent the exact values measured with a frequency counter); d is the $L^1$ distance between the measured and the exact values divided by the number of points that is the average discrepancy from the true frequency; the FSR is measured to be 15.15 GHz by minimizing the sum of d for the 4 Rb lines. (c) Suppression of Rayleigh-scattered light from a 150 mm long (orange line) and 75 mm long (blue line) Rb cell as a function of the temperature. The gray line represents the suppression of the two cells cascaded (d) Spectral resolution of the spectrometer, as measured from the FWHM of the laser line, after applying the reconstruction pipeline.

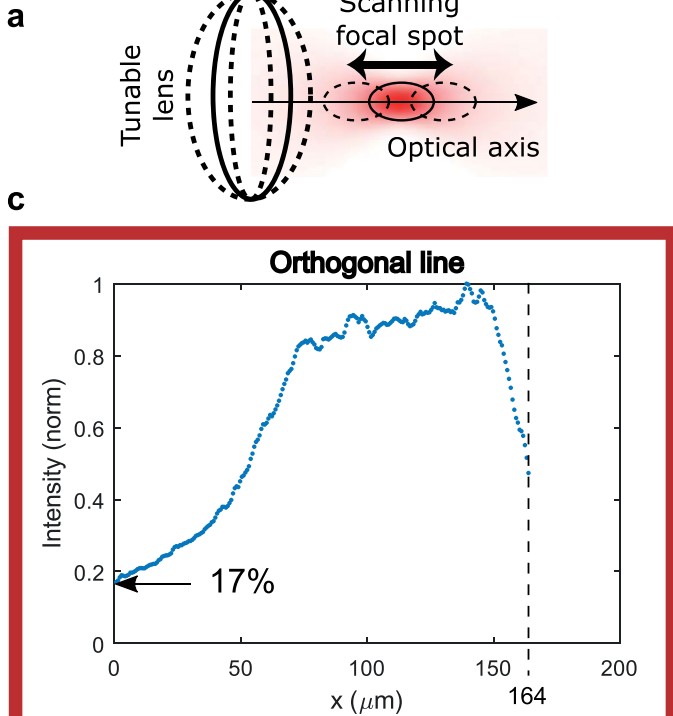

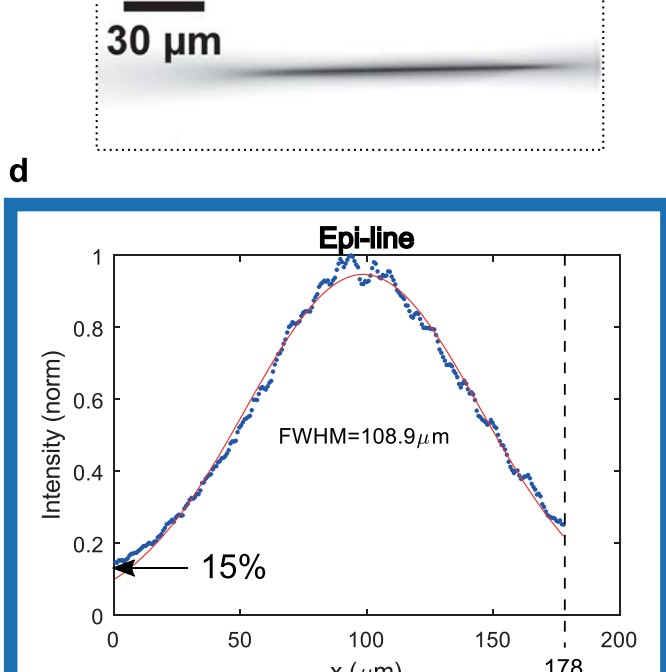

**Extended Data Fig. 6 | Characterization of the Brillouin illumination profile for both orthogonal and epi line geometries. (a)** Illustration of axial scanning by means of a tunable lens in order to extend the illumination depth and thus FOV in the orthogonal geometry. The curvature of the tunable lens can be controlled with an external voltage, thus shifting the focus spot in the sample. **(b)** Fluorescence image of the extended illumination line in the orthogonal geometry, acquired by imaging the fluorescence signal from ICG dye with a camera in the intermediate image plane (dashed line in **Extended Data** Fig. 3a). **(c, d)** Illumination profile along the FOV (measured as the intensity of the Stokes/ anti-Stokes spectral peaks in water) for the **(c)** orthogonal-line and **(d)** epi line geometries, respectively. Note that a relative intensity value of >15% is generally sufficient for reliable Brillouin peak fitting/detection.

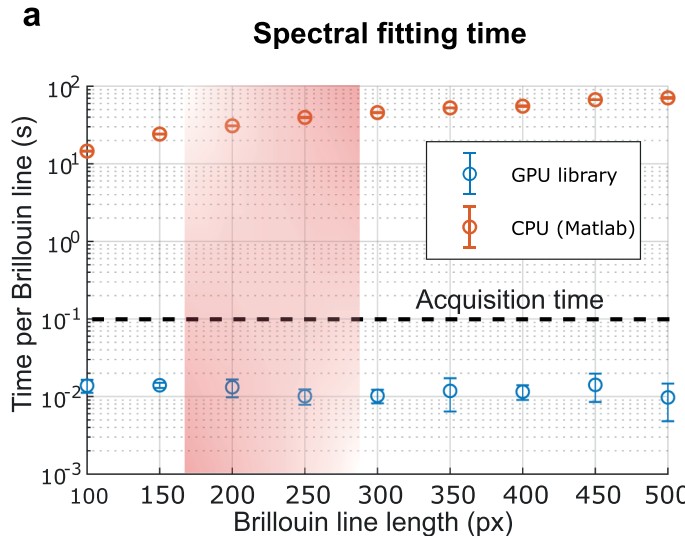

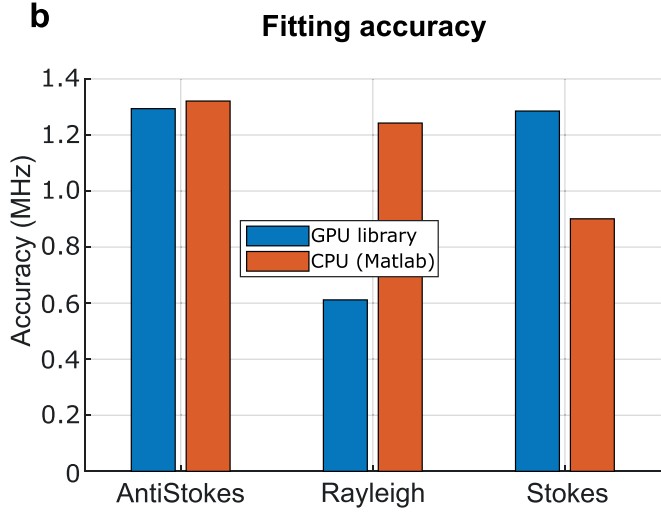

**Extended Data Fig. 7 | Performance of the GPU-accelerated spectral analysis pipeline. (a)** Brillouin spectrum fitting time as a function of image size (equal to number of independent spectra), comparing standard, CPU-based fitting routines (Matlab) and our custom GPU-accelerated library. The shaded red region shows the typical length of the Brillouin line in pixels. Error bars represent S.D. Note that most error bars are smaller than data points. **(b)** Fitting accuracy between the CPU- and GPU-based pipelines. Note that the accuracy uncertainty (~1 MHz) is negligible compared to the spectral fitting precision based on realistic, noisy data (~10–20 MHz).

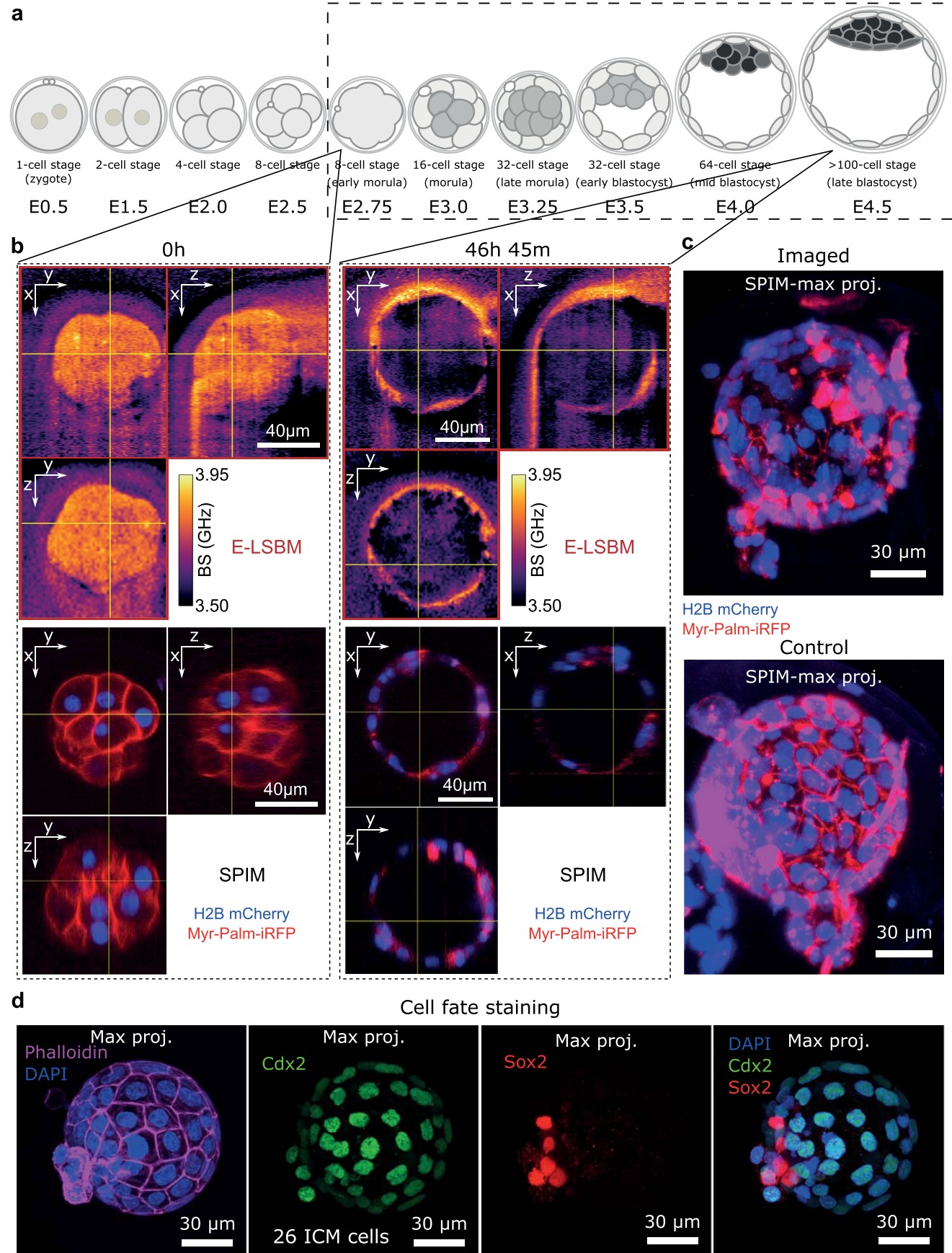

**Extended Data Fig. 8 | See next page for caption.**

**Extended Data Fig. 8 | LSBM allows Brillouin time-lapse imaging of developing mouse embryos over ~2 days. (a)** Timeline of mouse embryo development from the one cell stage to late blastocyst. The dashed rectangle encloses the developmental window that was imaged. **(b)** Exemplary Brillouin volumes acquired in the orthogonal geometry (top) and SPIM volumes (bottom) of a single mouse embryo at the beginning (left) and at end (46h45m after the first timepoint) of the time-lapse (right). The acquisition time for one volume in the Brillouin modality was ~11–17 min, the time interval between volumes is between 77 to 92 minutes. Representative result from n = 4 embryos. Also see Supplementary Videos 5, 6. **(c)** Top: SPIM volume (maximum intensity projection -MIP) of an embryo that underwent Brillouin time-lapse imaging, taken at the end of the time-lapse (representative image out of 4 embryos). Bottom: SPIM image (MIP) of a control embryo (taken at the same time as the embryo in top panel), that was in the same imaging chamber but not imaged by Brillouin or SPIM, showing qualitatively similar morphology (representative image out of 9 embryos). **(d)** MIPs through the volume where the outer trophectoderm cells are CDX2-positive (green), a marker for trophectoderm cell fate. The inner cell mass are SOX2-positive (red) and CDX2-negative, indicating proper epiblast fate in the ICM at late blastocyst stage. The number of cells in the ICM is 26, consistent with the reported values in literature [15]. Representative results from n = 3 embryos.

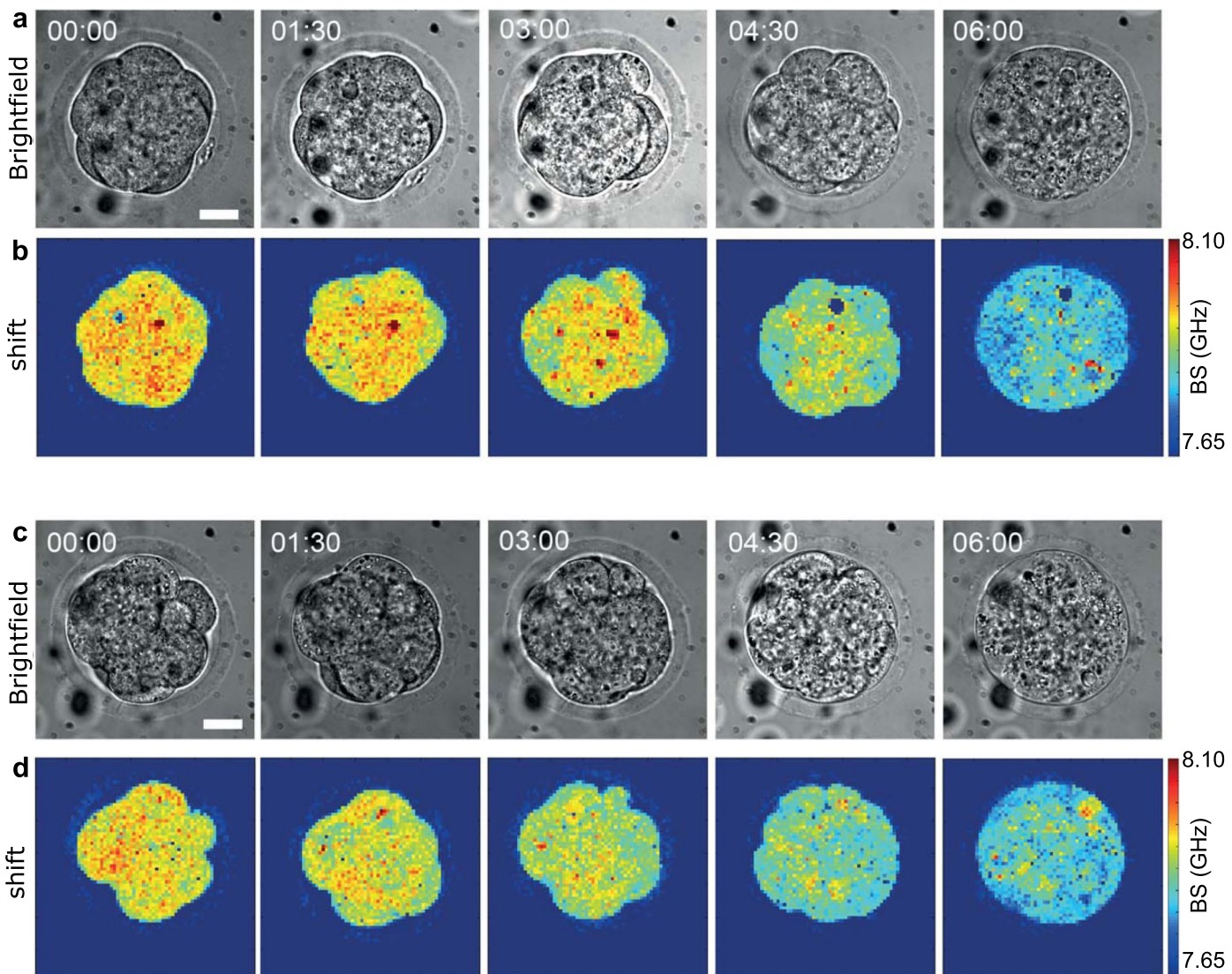

**Extended Data Fig. 9 | Embryos are not viable under 532-nm confocal Brillouin imaging. (a)** Bright-field images of an 8-cell stage mouse embryo that eventually undergoes cell death. **(b)** Images of Brillouin shift for the same embryo as in **a**. **(c)** Another representative embryo (out of 3 embryos in total) undergoing cell death under the same imaging condition. **(d)** Images of Brillouin shift for the same embryo as in **c**. Time is shown in h:mm. Scale bars = 20 μm.

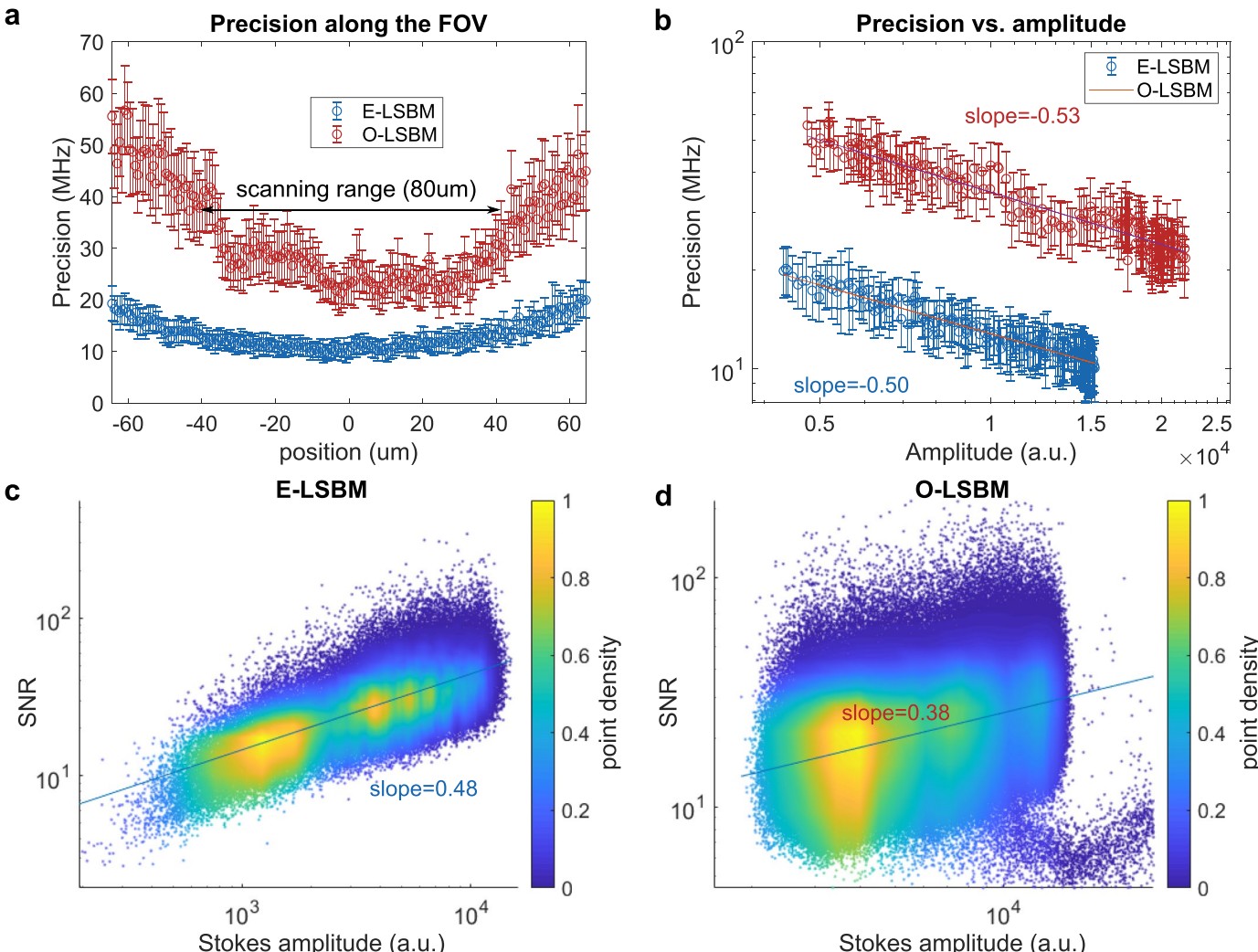

**Extended Data Fig. 10 | Performance characterization of LSBM. (a)** Analysis of the precision along the FOV for the O-LSBM (red) and E-LSBM (blue). **(b)** Analysis of the precision vs. amplitude along the FOV for the O-LSBM (red) and E-LSBM (blue). Data used for **(a-b)** were collected in water in typical imaging conditions, 100 ms exposure time and ~18 mW on the sample. The data points (circles) represent the precision, calculated as the S.D. of the Brillouin shift extracted from 441 subsequent spectra and the error bars are calculated as the S.D. of 20 different values of the precision, determined as described before and by grouping the 441 spectra in 22 groups of 20 each **(c-d)** Analysis of the SNR in a biological sample. For each spectrum the SNR is calculated as the amplitude of the Stokes peak divided by the standard deviation of the residuals (that is the difference between the raw data and the fit). The plot in **(c)** is generated from the dataset shown in Fig. 2c while **(d)** is from the dataset shown in Fig. 3b. The large variability in **(d)** could be attributed to the non-Lorentzian (that is broadened) [7] line shape of the O-LSBM modality.

# Reporting Summary

Nature Research wishes to improve the reproducibility of the work that we publish. This form provides structure for consistency and transparency in reporting. For further information on Nature Research policies, see Authors & Referees and the Editorial Policy Checklist.

## Statistical parameters

When statistical analyses are reported, confirm that the following items are present in the relevant location (e.g. figure legend, table legend, main text, or Methods section).

| n/a | Confirmed | |
|---|---|---|
| ☐ | ☒ | The exact sample size (*n*) for each experimental group/condition, given as a discrete number and unit of measurement |
| ☐ | ☒ | An indication of whether measurements were taken from distinct samples or whether the same sample was measured repeatedly |
| ☐ | ☒ | The statistical test(s) used AND whether they are one- or two-sided<br>*Only common tests should be described solely by name; describe more complex techniques in the Methods section.* |
| ☒ | ☐ | A description of all covariates tested |
| ☐ | ☒ | A description of any assumptions or corrections, such as tests of normality and adjustment for multiple comparisons |
| ☐ | ☒ | A full description of the statistics including central tendency (e.g. means) or other basic estimates (e.g. regression coefficient) AND variation (e.g. standard deviation) or associated estimates of uncertainty (e.g. confidence intervals) |
| ☐ | ☒ | For null hypothesis testing, the test statistic (e.g. *F*, *t*, *r*) with confidence intervals, effect sizes, degrees of freedom and *P* value noted<br>*Give P values as exact values whenever suitable.* |
| ☒ | ☐ | For Bayesian analysis, information on the choice of priors and Markov chain Monte Carlo settings |
| ☒ | ☐ | For hierarchical and complex designs, identification of the appropriate level for tests and full reporting of outcomes |
| ☒ | ☐ | Estimates of effect sizes (e.g. Cohen's *d*, Pearson's *r*), indicating how they were calculated |
| ☐ | ☒ | Clearly defined error bars<br>*State explicitly what error bars represent (e.g. SD, SE, CI)* |

*Our web collection on statistics for biologists may be useful.*

## Software and code

Policy information about availability of computer code

| Data collection | Custom written LabView 2018 control software as well as Arduino 1.8.13 code. |
|---|---|
| Data analysis | Custom written Matlab 2018a,b scripts, CUDA 11.5 code (see https://github.com/prevedel-lab/brillouin-gpu-acceleration), Fiji 1.52i, Prism Graphpad 8-9 |

For manuscripts utilizing custom algorithms or software that are central to the research but not yet described in published literature, software must be made available to editors/reviewers upon request. We strongly encourage code deposition in a community repository (e.g. GitHub). See the Nature Research guidelines for submitting code & software for further information.

## Data

Policy information about availability of data

All manuscripts must include a data availability statement. This statement should provide the following information, where applicable:
- Accession codes, unique identifiers, or web links for publicly available datasets
- A list of figures that have associated raw data
- A description of any restrictions on data availability

The raw datasets generated and/or analysed for this work are available at https://doi.org/10.5281/zenodo.7525851.

# Field-specific reporting

Please select the best fit for your research. If you are not sure, read the appropriate sections before making your selection.

☒ Life sciences ☐ Behavioural & social sciences ☐ Ecological, evolutionary & environmental sciences

For a reference copy of the document with all sections, see nature.com/authors/policies/ReportingSummary-flat.pdf

# Life sciences study design

All studies must disclose on these points even when the disclosure is negative.

| | |
|---|---|
| Sample size | This work focused on the development of a new imaging technique. The performance of the microscope was validated on imaging sub-diffraction (fluorescent) beads as well as live Drosophila and Phallusia embryos. The sample size was 64 which is the number of total imaging sessions and recordings, and was not based on any sample size calculation. In general, the imaging was repeated between 4 to 50 times on each organism, all of which produced comparable data quality. These numbers are indicated in the main text and below. This sample size was sufficient in our opinion since we were demonstrating a microscope's performance and were not investigating a biological question. |
| Data exclusions | No data was intentionally excluded from the study. Representative data sets were chosen for the Figures. |
| Replication | We repeated the in-vivo imaging experiments multiple times on a total of n=10 individual Drosophila and n= 60 Phallusia embryos and n=4 mouse embryos. The results were reproducible, i.e. they yielded image datasets of comparable quality. |
| Randomization | No randomization was applied as this was not essential for our study which concerned the demonstration of a new microscopy technique. |
| Blinding | In principle we were blinded to any group allocation and the outcome of our study, i.e. the demonstration of a new microscopy technique, is independent of the biological sample studied. |

# Reporting for specific materials, systems and methods

## Materials & experimental systems

| n/a | Involved in the study |
|---|---|
| ☒ | ☐ Unique biological materials |
| ☐ | ☒ Antibodies |
| ☒ | ☐ Eukaryotic cell lines |
| ☒ | ☐ Palaeontology |
| ☐ | ☒ Animals and other organisms |
| ☒ | ☐ Human research participants |

## Methods

| n/a | Involved in the study |
|---|---|
| ☒ | ☐ ChIP-seq |
| ☒ | ☐ Flow cytometry |
| ☒ | ☐ MRI-based neuroimaging |

# Antibodies

| | |
|---|---|
| Antibodies used | For Drosophila embryos: Anti Snail (rabbit) antibody against aminoacids 1-200 of Snail Protein, anti Drosophila Discs Large (DSHB, mouse, 4F3) and anti mCherry (Abcam rabbit, ab167453)<br><br>For mouse embryos: Primary antibodies against CDX2 (Biogenex, NC9471689; 1:150), SOX2 (Sigma, SAB3500187; 1:50) were used in this study. For the secondary staining, antibodies targeting mouse immunoglobulin coupled to Alexa Fluor 488 (Life Technologies, A21202), Alexa Fluor 555 (Life Technologies, A31570), rabbit immunoglobulin-coupled Alexa Fluor 546 (Invitrogen, A10040) were used. Finally, Alexa Fluor 633-coupled Phalloidin (Invitrogen, R415; 1:50) was used to stain filamentous Actin. |
| Validation | Snail: The antibody raised against the Snail-GST protein was characterised through immunostainings, immunoprecipitation and Western blot analysis. Snail validation information is available upon request.<br>Validation information about all commercial antibodies can be accessed on the corresponding supplier's websites. |

# Animals and other organisms

Policy information about studies involving animals; ARRIVE guidelines recommended for reporting animal research

| | |
|---|---|
| Laboratory animals | This work followed the European Communities Council Directive (2010/63/EU) to minimize animal pain and discomfort. All procedures described in this manuscript were approved by EMBL's committee for animal welfare and institutional animal care and use (IACUC), under protocol number 2020-01-06RP. (C57BL/6xC3H) F1 mice from eight-weeks of age onwards were used. |

Embryos were recovered from superovulated female mice mated with male mice.
Drosophila are invertebrate organisms and thus do not require ethics oversight.

Wild animals

Adult wild Phallusia mammillata were collected by the Roscoff Marine Biological Station (France) in the coast of Brittany. The Phallusia were shipped from Roscoff to Heidelberg where they were maintained in an aquarium until used. The gametes were surgically collected for in vitro fertilisation and the sacrificed Phallusia killed by freezing. Phallusia are invertebrate organisms and thus do not require ethics oversight.

Field-collected samples

The study did not involve field-collected samples.

