## [Peer Review File · Nature Methods]

Peer Review Information

Manuscript Title: High-resolution line-scan Brillouin microscopy for live-imaging of mechanical properties during embryo development

Corresponding author name(s): Robert Prevedel

Editorial Notes: n/a

Reviewer Comments & Decisions:

Decision Letter, initial version:

Dear Robert,

Thank you for your patience. Your Brief Communication, "High-resolution line-scan Brillouin microscopy for live-imaging of mechanical properties during embryo development", has now been seen by three reviewers. As you will see from their comments below, although the reviewers find your work of considerable potential interest, they have raised a number of important concerns. We are interested in the possibility of publishing your paper in Nature Methods, but would like to consider your response to these concerns before we reach a final decision on publication.

We therefore invite you to revise your manuscript to address these concerns. I strongly recommend that you discuss a revision plan with me before embarking on any experiments.

* include a point-by-point response to the reviewers and to any editorial suggestions

* please underline/highlight any additions to the text or areas with other significant changes to facilitate review of the revised manuscript

* address the points listed described below to conform to our open science requirements

* ensure it complies with our general format requirements as set out in our guide to authors at www.nature.com/naturemethods

* resubmit all the necessary files electronically by using the link below to access your home page

[Redacted] This URL links to your confidential home page and associated information about manuscripts you may have submitted, or that you are reviewing for us. If you wish to forward this email to co-authors, please delete the link to your homepage.

We hope to receive your revised paper within 2 months. If you cannot send it within this time, please let us know. In this event, we will still be happy to reconsider your paper at a later date so long as nothing similar has been accepted for publication at Nature Methods or published elsewhere.

OPEN SCIENCE REQUIREMENTS

REPORTING SUMMARY AND EDITORIAL POLICY CHECKLISTS

Please note that these forms are dynamic 'smart pdfs' and must therefore be downloaded and completed in Adobe Reader. We will then flatten them for ease of use by the reviewers. If you would

like to reference the guidance text as you complete the template, please access these flattened versions at <http://www.nature.com/authors/policies/availability.html>.

DATA AVAILABILITY

All novel DNA and RNA sequencing data, protein sequences, genetic polymorphisms, linked genotype and phenotype data, gene expression data, macromolecular structures, and proteomics data must be deposited in a publicly accessible database, and accession codes and associated hyperlinks must be provided in the “Data Availability” section.

Please include a “Data availability” subsection in the Online Methods. This section should inform readers about the availability of the data used to support the conclusions of your study, including accession codes to public repositories, references to source data that may be published alongside the paper, unique identifiers such as URLs to data repository entries, or data set DOIs, and any other statement about data availability. At a minimum, you should include the following statement: “The data that support the findings of this study are available from the corresponding author upon request”, describing which data is available upon request and mentioning any restrictions on availability. If DOIs are

provided, please include these in the Reference list (authors, title, publisher (repository name), identifier, year). For more guidance on how to write this section please see:
<http://www.nature.com/authors/policies/data/data-availability-statements-data-citations.pdf>

CODE AVAILABILITY

Please include a “Code Availability” subsection in the Online Methods which details how your custom code is made available. Only in rare cases (where code is not central to the main conclusions of the paper) is the statement “available upon request” allowed (and reasons should be specified).

For more information on our code sharing policy and requirements, please see:
<https://www.nature.com/nature-research/editorial-policies/reporting-standards#availability-of-computer-code>

MATERIALS AVAILABILITY

ORCID

Nature Methods is committed to improving transparency in authorship. As part of our efforts in this direction, we are now requesting that all authors identified as ‘corresponding author’ on published papers create and link their Open Researcher and Contributor Identifier (ORCID) with their account on the Manuscript Tracking System (MTS), prior to acceptance. This applies to primary research papers only. ORCID helps the scientific community achieve unambiguous attribution of all scholarly contributions. You can create and link your ORCID from the home page of the MTS by clicking on

'Modify my Springer Nature account'. For more information please visit www.springernature.com/orcid.

Best regards,
Nina

Nina Vogt, PhD
Senior Editor
Nature Methods

Reviewers' Comments:

Reviewer #1:

Remarks to the Author:

The authors present a 780-nm line-scanning Brillouin microscope (LSBM) and demonstrate imaging of dynamical processes in living samples with low-phototoxicity. Although the developed microscope is technically impressive, it appears that LSBM cannot stand as a robust method on its own as simultaneous fluorescence guidance is necessary (because the Brillouin contrast is rather low as observed in all the LSBM images). While the need for fluorescence guidance may be acceptable though it will probably hinder the widespread use of the method, the spatial details provided by the Brillouin contrast are weak or absent, particularly when considering the corresponding spatial details of the fluorescence images or the high-resolution confocal Brillouin images of living organisms (e.g., <https://doi.org/10.1364/BOE.10.001420>). Due to the lack of spatial details, the quantification of the LSBM data is summarized by relative spatial means of the Brillouin frequency shift over large regions, which is not suitable for the high-resolution analysis of dynamic processes over space and time in living organisms. I am unable to recommend publication without the authors addressing the comments/questions below.

1. The definition of SNR should clearly be presented.

2. The phonons should be sketched with different size in the O-LSBM and E-LSBM schemes (Figs. 1c and 1d). How does the different size of the phonons and the spectral broadening of these two scattering geometries affect the Brillouin imaging resolution? These are important issues to discuss.
3. Were there artifacts in the Brillouin measurements owing to the single line illumination utilised?
4. Why are there wings in the O-LSBM PSF? Do the solid lines represent theoretical or numerical fits? This should clearly be indicated.
5. What are the factors that define the size of the focused line? How do the size of the focused line and the experimental spectral resolution of the instrument affect the mechanical resolution of LSBM? It is important to discuss this matter carefully.
6. The precision of E-LSBM seems to be better than that of O-LSBM. Is this a fundamental difference between the two scattering geometries? Why was the precision of E-LSBM measured at 532 nm? A good comparison requires the use of the same wavelength. The relative precision measured for E-LSBM seems to be twice as high as that measured for O-LSBM. How does this difference affect the ability to identify cellular and subcellular components? It is crucial to discuss all these issues, particularly in light of the work reported in <https://doi.org/10.1364/BOE.10.001567>.
7. Why is only the coarse structural mechanical dynamics detected by LSBM in Figs. 2c and 2f? Is it possible to significantly improve the spatial and mechanical resolution? Is there a fundamental limit to the spatial and mechanical resolution of LSBM?
8. Would an analysis of the spatial variance of the Brillouin frequency shift across the regions of interest in Figs. 2c and 2f result in additional insights about the investigated dynamics?
9. The scale bar in Figs. 2c and 2f is not clear and should be corrected.
10. What is the scale for sub-cellular components in the *Phallusia mammillata*? Can the spatial details in Figs. 3b and 3e be significantly improved by increasing the effective pixel time (to increase the SNR) and the spatial sampling/resolution?
11. Is the data in Fig. 3j statistically significant? Can the three Brillouin shift frequency components be identified distinctly from the entire image without segmentation? Why?
12. Were the Brillouin signals detected when imaging the living organisms also shot noise limited, as measured in water? According to which criteria was the effective pixel time of 1 ms chosen in the biological samples?
13. The time interval for calculating the illumination energy in SI Fig. 4d should be mentioned.
14. Characterization of the precision and accuracy of the LSBM spectrometer as a function of the spatial location in water is important (e.g., the interference between the spatial and spectral dimensions on the camera).
15. Why are the nuclei weakly/not detected by LSBM in SI Fig. 8b? Will a larger effective pixel time help? Can a confocal Brillouin microscope detect them safely? A detailed comparison between the imaging performance of confocal Brillouin microscopy and LSBM is required.
16. What will be the results of the experiment of SI Fig. 9 using 780-nm confocal Brillouin imaging? Where will phototoxicity-optimized 780-nm confocal Brillouin microscopy appear in SI Fig. 4d? will it overlap the operation region of LSBM?

17. Can LSBM with an effective pixel time of 1 ms pixel be shown to achieve high-resolution images of living organisms comparable to those of confocal Brillouin microscopy (e.g., <https://doi.org/10.1364/BOE.10.001420>)?
18. What is the maximum frame rate of the LSBM instrument?
19. If the analysis of the LSBM data comes down to the mean Brillouin frequency shift over the image (or over a few large, segmented regions of the image), why would not time lapsed imaging of a line in the sample by LSBM or multiple points in the sample by a 780-nm confocal Brillouin microscope with fluorescence guidance be adequate to study the samples presented in this work? This would enable to measure faster dynamics limited by the camera exposure time rather than by the volume imaging time of LSBM. Nevertheless, it should explicitly be noted in the manuscript that the fastest dynamics LSBM can probe is limited by the camera exposure time of ~100 ms rather than the effective pixel time.
20. Are 3 embryos an adequate sample size for the statistical analysis presented in Figs. 2d and 3j?

Reviewer #2:

Remarks to the Author:

A Summary of Key Results: The Brillouin microscope reported here uses line scanning and epi or orthogonal detection to increase image speed to circa 1 ms per voxel whilst achieving up to 180x165x170 μ m field-of-view (FOV) with down to 1.5 μ m spatial and down to 2min temporal resolution, whilst achieving circa 80 dB spectral extinction and a spectral precision of <20 MHz. The microscope also incorporates a SPIM setup for co-registered fluorescence imaging.

The key results in this work are three examples to show that Brillouin microscopy's capability in relation to temporal dynamics (live drosophila embryo), high-mechanical property resolution (the ascidian *Phallusia mammillata*), and low photo-toxicity (mouse embryo). I would have found a table summarising the experiment, including the mode and variable FOV and spatial resolution to be very helpful in sorting out the myriad details. Could this be included?

In epi mode at 20 mW illumination power is viable (no observed photodamage and 24-hr post viability) on live drosophila embryo over a time lapse of 30 minutes or so, at around 10 times lower illumination energy per pixel. Co-registered SPIM is included.

In orthogonal mode, the microscope was used to image the ascidian *Phallusia mammillata* over up to 14 hours – what was the illumination power of the laser? Co-registered SPIM is included.

And (can the mode be stated please – paragraph commencing line 172) with acquisition times of ~11-17min at 75-90min intervals over 46 hours, sensitive mouse embryos were shown to be undamaged, whereas, damaged when imaged using “conventional” confocal imaging parameters and wavelength.

B Originality and significance: Line scanning in an orthogonal detection geometry was proposed and published in 2016 – Sci Reports: Ref 20. Can explicit reference be made to the novelty in the current manuscript? I expect this may be a collection of incremental improvements that collectively add up to a technically challenging and high-performance implementation – relative to previous incarnations, but this should be clearer from the manuscript. If the significance is to claim new mechanobiological capability, then this too should be clearly stated. Right now, it seems this manuscript would be well suited for Reviews of Scientific Instruments – a complex and highly capable instrument developed and described with exemplar samples and targets. The claimed advances need to be more specifically made and substantiated.

In lines 189-191, in summary, it is claimed “Compared to alternative Brillouin scattering approaches and implementations^{7,9,13,14,18} this represents a >20-fold improvement in terms of imaging speed, at >10-fold lower illumination energy per pixel without sacrifices in measurement precision⁷.” This is very clear, but it is not clear which specification is demonstrated by which experiment, i.e., whether they can be achieved simultaneously, and the claim of high mechanical property resolution of example 2, appears to be contradicted by the claim of without sacrificing measurement precision, when an improvement might have been expected.

Overall, it is indeed encouraging to see Brillouin microscopy continue to progress as a non-contact mechanobiological imaging technique – I am supportive of publication, once minor issues are addressed and subject to a convincing explanation of novelty and significance.

C Data & methodology: There is a huge amount of detail in methods and supplementary information that broadly speaking is well presented and convincing. As indicated, a table summarising the three examples would help as there is a lot of change between experiments.

Can the authors spell out any negative consequences of incorporating SPIM, for example, on photo-toxicity?

For the orthogonal mode, it would seem that there is potential for droop in Brillouin frequency versus illumination depth, arising from refraction, sample heterogeneity, and the Brillouin frequency shift's dependence on angle to the incident beam. Was this observed? If so, what was its magnitude? If not, why not?

D Appropriate use of statistics: No specific comments.

E Conclusions: Overall, this manuscript provides a comprehensive description of the instrument and the experiments performed using it, but the wood is lost for some extent to the trees – the advances in capability are less clear. See above for novelty and significance. As regards the experimental findings, can the conclusion/summary be clear on what has been seen before and what is reported here for the first time.

F Suggested improvements: experiments, data for possible revision: As above, greater clarity around what the novelty and significance is would help frame the manuscript as a pivotal one whereas at present this clarity is lost in the detail.

G References: Appropriate credit is given to previous work.

H Clarity and context: I found the abstract rather weak and did not convey the key novelty and achievements well at all.

Reviewer #3:

Remarks to the Author:

The manuscript “High-resolution line-scan Brillouin microscopy for live-imaging of mechanical properties during embryo development” by Bevilacqua et al propose a new Brillouin microscopy design, which brings a much-needed improvement to the technique. Indeed, combining line-scanning approach with near infra-red illumination, their proposed design allows for a faster less-toxic acquisition of Brillouin microscopy images without sacrificing measurement precision, a quality that is important for long term imaging live samples and especially dynamic processes. Furthermore, the microscope has a dual mode: an O-LSBM to allow for better axial resolution and lower phototoxicity, and an E-LSBM that minimises effects from scattering and optical aberrations (at the cost of lower axial resolution) and thus is suited for heterogenous samples. Finally, adding a GPU-accelerated routine for data visualization makes the microscopy technique more practical and insightful to the user, as it allows for in situ visualization of the data. The authors test their microscope in three different model organisms, exploring the potential applications of their system.

The manuscript is written clearly, and the appropriate controls and measures are conducted in most places and described in detail. There are however a few comments and suggestions that I believe would

improve the quality of the manuscript. Overall, I believe that this manuscript will open the way for increased use of Brillouin microscopy in biological/biomedical studies, and would therefore recommend publication in Nature Methods following consideration of the points below.

Main comments/suggestions:

Line 64-66: "Our microscope is based on a line-scanning approach that enables multiplexed signal acquisition, allowing the simultaneous sensing of hundreds of points and their spectra in parallel."

Considering that 90° scattering geometry of LSBM broadens spectra and therefore limits resolution¹, could the authors please comment on if/how they have overcome this limit?

E-LSBM vs O-LSBM: From Fig 1f, it seems that the Brillouin Shifts measured by the E-LSBM are systematically higher than those measured by O-LSBM. Is this something expected from the system (e.g. from the optical geometry/setup). Furthermore, Fig S14d shows that for the same illumination energy, the precision of O-LSBM is more than E-LSBM, hence being less toxic for live imaging. However this contradicts with Fig 1f, where it seems that although O-LSBM gives better spatial resolution, the precision of the E-LSBM is higher (smaller standard deviation). Could the authors please clarify this, for example perhaps different laser power/illumination energy was used when comparing the two modes in Fig 1f?

Line 144-146: "Similar to the observations during VFF, the average Brillouin shift within cells engaged in tissue folding also increased during PM". The cells in the contractile region are probably those that move and deform most. I was wondering if the movement of cells could affect the light scattering and therefore the Brillouin shift? If so, then the increased Brillouin shift in this region might not necessarily be the result of the higher contractility but the result of higher movement? Could the authors please comment on this.

Line 147-149: "No photodamage or -toxicity was observed at <~20mW of average laser power, and viability assays showed that all embryos (n=3) imaged progressed to the first larval stage (24hpf)." The fact that all imaged embryos progressed to the first larval stage is indeed a strong indication of minimal photodamage. However, the authors could make their case even stronger if they showed that the imaged embryos developed into normal adults.

Line 157-160: "We observed a perinuclearly localized, high Brillouin signal within the B5.2 cells in the late 16-cell stage (Fig. 3e,f). This subcellular region is known to have a dense microtubule bundle structure driven by the centrosome attracting body (CAB) (Fig. 3d)." The authors use this example to show the ability of the O-LSBM to mechanically probe subcellular structures with high resolution, and this would suffice for the current manuscript. However, their result and manuscript would be much stronger if they could perturb the microtubule structure (for example with Nocodazole treatment) and

show that the stiff region is gone, confirming that the high Brillouin shift region is indeed due to the dense microtubule bundle structure in these cells.

Fig 3f: It seems that the authors have Brillouin microscopy images of 2 organisms (caption of figure 3b,c). It would therefore be beneficial to plot the Brillouin shift along a similar line in the 2nd organism to show reproducibility.

Line 167-169: “These experiments also demonstrate that LSBM imaging can be used over long periods (here 14 hours; SI Video 3-4)”. The authors indeed show that the organism is viable over long period of imaging. However, a more quantitative analysis (e.g. LIVE/DEAD assay or similar methods) would be desirable, especially because low phototoxicity is one of the main features of the proposed microscopy method.

Line 172-174: “Finally, to test the low-photo-toxicity of the LSBM approach in a further organism, we imaged the developing mouse embryo, from the 8-cell stage (E2.75) to the late-blastocyst (E4.5), covering a 46-hour time-span (SI Fig. 8).” The authors only show the initial and final timepoints of this experiment. It would be beneficial to include a movie of this process so the reader could see the intermediate timepoints as well.

Line 177-180: “Despite the embryos’ notorious photo-sensitivity, no photodamage or -toxicity was observed at $< \sim 20$ mW of average laser power as confirmed by the morphology, dynamics, cell number and cell fate of the imaged embryos resembling those of control embryos (SI Fig. 8c,d).” Currently the authors show exemplary images of the imaged embryos and show that they are normal in terms of morphology and cell fate. It would be beneficial to include more quantitative measures of cell number and dynamics to confirm that the long-term imaging was non-toxic.

Minor comments:

Line 324: Typo. “raise” should be “rise”.

Fig 1e: Do the circle, cross and star represent different experiments? Also for both Fig 1e and f, it would be beneficial to include a more detailed description of the measurements and how they should be interpreted in the caption.

Fig 3j: What do the inner error bars represent?

Fig S4 title: Typo. “BLSM” should probably be “LSBM” to be consistent with the rest of the manuscript.

Fig S4d, S7a: Please mention what the error bars represent.

Fig S8c,d: Is the n number the same as Fig S8a, i.e. $n=4$?

Fig S9b,d: Numbers on the colorbar are hard to read.

References:

1 Prevedel, R., Diz-Muñoz, A., Ruocco, G. & Antonacci, G. Brillouin microscopy: an emerging tool for mechanobiology. *Nature Methods* 16, 969-977, doi:10.1038/s41592-019-0543-3 (2019).

Author Rebuttal to Initial comments

Point-by-point reply for NMETH-BC48320**Remark to all Reviewers:**

We would like to thank all Reviewers for their reports and valuable comments, which we believe, have helped to significantly enhance the quality of our manuscript. Below we give a point-by-point response to all issues that were raised and how we have addressed them in the revised version of our manuscript. To facilitate review of the revised manuscript, we have underlined any additions to the text or areas with other significant changes. We are confident that these improvements should fully address all Reviewers' comments and suggestions, and hope that it now meets their expectations.

Original Reviewer comments are in black. Our replies are in blue.

Reviewers' Comments:**Reviewer #1:****Remarks to the Author:**

The authors present a 780-nm line-scanning Brillouin microscope (LSBM) and demonstrate imaging of dynamical processes in living samples with low-phototoxicity. Although the developed microscope is technically impressive, it appears that LSBM cannot stand as a robust method on its own as simultaneous fluorescence guidance is necessary (because the Brillouin contrast is rather low as observed in all the LSBM images). While the need for fluorescence guidance may be acceptable though it will probably hinder the widespread use of the method, the spatial details provided by the Brillouin contrast are weak or absent, particularly when considering the corresponding spatial details of the fluorescence images or the high-resolution confocal Brillouin images of living organisms (e.g., <https://doi.org/10.1364/BOE.10.001420>). Due to the lack of spatial details, the quantification of the LSBM data is summarized by relative spatial means of the Brillouin frequency shift over large regions, which is not suitable for the high-resolution analysis of dynamic processes over space and time in living organisms. I am unable to recommend publication without the authors addressing the comments/questions below.

We thank the Reviewer for their detailed comments and questions which made us realize that important aspects of our work have not been clearly communicated. During the revision we have added the missing technical details and discussions in the form of additional SI material (1 SI Note and 1 SI Fig.).

Regarding the above comments that our Brillouin method 'cannot stand as a robust method on its own as simultaneous fluorescence guidance is necessary' and 'Brillouin contrast are weak or absent', we would like to point out that in Brillouin shift images, the pixel value reports on the mechanical property (elasticity), and thus a 'low contrast' is not an indication for the method not working properly, but rather indicates that there does not seem to be any spatial differences in mechanics. Our motivation for integrating a fluorescence (SPIM) modality was to be able to perform spatially resolved analysis informed by molecular constituents, e.g. in a cell- or tissue-specific manner, and therefore to help in the analysis and interpretation of the acquired images (see original Fig. 2b,c and 3g-j). In other words, unless the mechanical properties correlate perfectly with a particular cellular compartment (e.g.: cytoplasm, nucleus, etc.), which is rather unusual and cannot be expected a priori, fluorescent microscopy becomes an additional asset for the interpretation of Brillouin shifts. We now specifically comment on this in the revised manuscript and in a separate SI Note section (*Fluorescence SPIM modality assists quantification of Brillouin images*).

1. The definition of SNR should clearly be presented.

We thank the Reviewer for pointing out the missing definition of SNR that could lead to ambiguity due to the multiple existing definitions. The reported SNR is the ratio between the amplitude of the Stokes peak divided by the standard deviation of the residuals after fitting the spectra with the appropriate function (i.e. the difference between the raw data and the fit. We have added this definition to the Methods section (*LSBM system characterization and image acquisition*) of the manuscript.

2. The phonons should be sketched with different size in the O-LSBM and E-LSBM schemes (Figs. 1c and 1d). How does the different size of the phonons and the spectral broadening of these two scattering geometries affect the Brillouin imaging resolution? These are important issues to discuss.

In the manuscript we mainly discussed the optical resolution which is determined by the microscope optics, while the “Brillouin” resolution depends also on the sample in a non-trivial manner (Caponi, Opt. Lett. 2020, <https://opg.optica.org/ol/abstract.cfm?uri=ol-45-5-1063>). We agree with the Reviewer that a discussion about the difference in “Brillouin” resolution (both spatial and spectral) between the two geometries would be informative and have thus added a dedicated Supplementary Note to the manuscript. We further thank the Reviewer for spotting that the phonons in the two geometries should have different sizes and have updated Figure 1 accordingly.

3. Were there artifacts in the Brillouin measurements owing to the single line illumination utilised?

We are not sure whether the Reviewer is referring to a comparison between “standard” point-scanning confocal implementation and ours or referring to the fact that absorption and refraction of the illumination line in the O-LSBM can degrade the image quality. In the former case we elaborate more on this in our reply to question 17. In the latter case, we recognised that this can be an issue for highly heterogeneous samples, which is also shared with fluorescence lightsheet microscopy. This is why we developed the E-LSBM configuration to overcome this limitation. Another possible strategy (also used in fluorescence lightsheet microscopy), is to have a dual illumination (as also recently demonstrated by the Zhang/Scarcelli group). We now discuss these points in more detail in the new SI Note (*Effect of refraction on the LSBM signal*).

4. Why are there wings in the O-LSBM PSF? Do the solid lines represent theoretical or numerical fits? This should clearly be indicated.

Since the z resolution of the objective is larger than the “thickness” of the illumination light in the O-LSBM configuration, the wings in the PSF are likely due to the convolution between the optical PSF of the objective and the illumination profile, probably combined with non-perfect focusing of the illumination line. The solid line is a numerical Gaussian fit, used to determine the FWHM. We have added this information to the Figure caption.

5. What are the factors that define the size of the focused line? How do the size of the focused line and the experimental spectral resolution of the instrument affect the mechanical resolution of LSBM? It is important to discuss this matter carefully.

In the E-LSBM configuration, the size of the line and the z resolution are coupled, so the full NA of the objective must be used to obtain the best possible z resolution. In the O-LSBM the size of the illumination line can be freely tuned by changing the effective NA of (i.e. the beam size before) the illumination objective. We designed the size of the illumination line to be below $\sim 2\mu\text{m}$, that is the typical mechanical resolution in biological samples.

As already mentioned in the reply to question 2, we now discuss how the optical PSF and the spectral resolution affect the mechanical resolution in a new Supplementary Note (*Comparison of spatial resolution in O-LSBM vs. E-LSBM*).

6. The precision of E-LSBM seems to be better than that of O-LSBM. Is this a fundamental difference between the two scattering geometries? Why was the precision of E-LSBM measured at 532 nm? A good comparison requires the use of the same wavelength. The relative precision measured for E-LSBM seems to be twice as high as that measured for O-LSBM. How does this difference affect the ability to identify cellular and subcellular components? It is crucial to discuss all these issues, particularly in light of the work reported in <https://doi.org/10.1364/BOE.10.001567>.

The reason for this difference is twofold: First, the 90deg geometry has lower collection efficiency due to the mismatch between the illumination and detection NA (see Zhang 2016 and our SI Note - *Efficiency for O-LSBM vs. E-LSBM* - for a detailed calculation); second, the Brillouin peaks are significantly broadened at 90 deg (Antonacci 2013) and, as it is well known from localisation microscopy, the precision of localisation depends on the width of the distribution. We have added a discussion about this in the SI Note.

Indeed, the lower precision might affect the ability to visually identify structures if the difference between their Brillouin shift and the surrounding pixels falls below the precision. But, if the aim of the experiment is to look at some specific structure, one can segment the structure of interest from the fluorescence channel and average the pixels belonging to it, thus gaining more statistical power to actually distinguish it. This again highlights the usefulness and power of the additional fluorescence SPIM modality.

Finally, we want to point out that the precision for both the E-LSBM and O-LSBM configuration were actually measured at 780nm, not 532nm.

7. Why is only the coarse structural mechanical dynamics detected by LSBM in Figs. 2c and 2f? Is it possible to significantly improve the spatial and mechanical resolution? Is there a fundamental limit to the spatial and mechanical resolution of LSBM?

As stated above, we have expanded on the discussion about the resolution in Brillouin microscopy, the limitations to further improve it, and the differences between our modalities in a SI Note.

8. Would an analysis of the spatial variance of the Brillouin frequency shift across the regions of interest in Figs. 2c and 2f result in additional insights about the investigated dynamics?

This is an interesting comment but we are unsure if this is meant to go beyond what we are already providing in Fig. 2d. The standard deviation at each time point plotted there actually represents precisely how much variation around the mean of the region of interest we detected across 3 projections for each embryo (see Methods Section - *Drosophila embryo imaging*). However, the most informative source of spatial variance are the reported images themselves (Fig. 2b,e) which are representative of the 3 independent experiments. In these images (and the Supplementary Video) one can appreciate not only the different sizes of Brillouin shifts co-existing in the region of interest but also where they were detected (aided by the fluorescence SPIM modality).

9. The scale bar in Figs. 2c and 2f is not clear and should be corrected.

We thank the Reviewer for pointing this out and have corrected the scale bars accordingly.

10. What is the scale for sub-cellular components in the *Phallusia mammillata*? Can the spatial details in Figs. 3b and 3e be significantly improved by increasing the effective pixel time (to increase the SNR) and the spatial sampling/resolution?

Cytoskeletal components such as microtubule bundles can range in size and length in *Phallusia*, but are typically on the order of $\sim\mu\text{m}$. While an increase in pixel time would indeed likely increase the SNR, this would entail too long acquisition times (for the whole volume) which would cause problems in terms of imaging artefacts, since the animal is rapidly developing (and thus moving). It was indeed the main motivation to show that our LSBM is sufficiently fast to capture an entire volume before the animal divides further (on a timescale of $\sim 20\text{min}$). One could decrease the spatial sampling, but this would lead to further loss of spatial detail, thus making sub-cellular scale ($\sim\mu\text{m}$) imaging difficult. We performed a careful optimization of all parameters and used the set that represents the best possible trade-off between resolution, sampling and volume speed. We have added a discussion to this point in the new SI Note (*Acquisition time of the LSBM*).

11. Is the data in Fig. 3j statistically significance? Can the three Brillouin shift frequency components be identified distinctly from the entire image without segmentation? Why?

We agree with the Reviewer that a statistical test would be informative. Performing a statistical analysis on the original Fig. 3j, as suggested by the Reviewer, would indeed show a statistically significant difference between the pixel distributions of the three tissue domains within a single embryo. However, we realize this could give an (biological) impression that this is a general feature of the analysed tissues in *Phallusia*. Therefore, we decided to change the panel to show the data across all imaged embryos ($n=3$) – see new Fig.3h. Applying a one-way ANOVA, paired, non-parametric test (Friedman Test) to this data we found no statistically significance ($p = 0.1944$). Likely, more data is required to conclusively answer this biological question, which we however find is outside the scope of the current study. In our work, we wanted to highlight that it is possible to perform cell and tissue specific Brillouin analysis in 3D, enabled by our LSBM approach that permits the fast acquisition of volumetric Brillouin data.

We further note that it is difficult to distinguish the different tissues directly from the Brillouin image without the fluorescence segmentation. This highlights again the usefulness of the SPIM modalities which allows to quantify the average Brillouin shifts in a cell and tissue specific measurements without relying on the mechanical contrast alone.

12. Were the Brillouin signals detected when imaging the living organisms also shot noise limited, as measured in water? According to which criteria was the effective pixel time of 1 ms chosen in the biological samples?

We have analysed our images to see whether they are indeed shot-noise limited and included the outcome as a new SI Fig 10c,d. Indeed, our analysis of the SNR vs. signal amplitude of all pixels within the biological samples show a slope close to ~ 0.5 , demonstrating shot noise limited performance also in sample regions with lower signal intensity.

Fig. R1: Analysis of the SNR in a biological sample. For each spectrum the SNR is calculated as the amplitude of the Stokes peak divided by the standard deviation of the residuals (i.e. the difference between the raw data and the fit). The plot in (c) is generated from the dataset shown in Fig. 2c while (d) is from the dataset shown in Fig. 3b. See new SI Fig. 10c,d.

The pixel time was chosen to be short enough to provide sufficient speed to capture 3D and dynamics. At the same time a camera exposure of 100ms is the lowest time that is significantly higher than the dead time between subsequent camera acquisitions (in the order of tens of ms, due to the camera readout and stage movement). We have added this to the new SI Note (*Acquisition time of the LSBM*).

13. The time interval for calculating the illumination energy in SI Fig. 4d should be mentioned.

The time intervals were 70ms for the two points with the lowest illumination energy, 200ms for the point with highest illumination energy and 100ms for the rest. However, we note that the precision depends only on the energy (power x time), which is stated in the x-axis, and not on the power or time interval alone.

14. Characterization of the precision and accuracy of the LSBM spectrometer as a function of the spatial location in water is important (e.g., the interference between the spatial and spectral dimensions on the camera).

We agree that such analysis is informative and have added these plots as a new Supplementary Figure 10a,b. In short, we find that the precision is <30MHz and <15MHz within the central ~100µm FOV for the O-LSBM and E-LSBM, respectively.

Fig. R2: Performance characterization of LSBM. (a) Analysis of the precision along the FOV for the O-LSBM (red) and E-LSBM (blue). In O-LSBM the precision is quickly degrading outside of the range

covered by the scanning of the tunable lens due to the decrease of the intensity. (b) Analysis of the precision vs. amplitude along the FOV for the O-LSBM (red) and E-LSBM (blue). The slope of -0.5 shows that the precision is shot noise limited even in the regions of the FOV with lower signal intensity. (Data used for (a-b) were collected in water in typical imaging conditions, 100ms exposure time and $\sim 18\text{mW}$ on the sample). See new SI Fig. 10a,b.

We further note that the characterization of the accuracy over the FOV was already included in the original manuscript as SI Fig. 5b.

15. Why are the nuclei weakly/not detected by LSBM in SI Fig. 8b? Will a larger effective pixel time help? Can a confocal Brillouin microscope detect them safely? A detailed comparison between the imaging performance of confocal Brillouin microscopy and LSBM is required.

Nuclei do not necessarily always display distinct mechanical properties. In fact, the nuclei are also barely distinguishable in SI Figure 9 which shows mouse embryos acquired with the 532nm confocal. On the other hand, nucleoli are quite pronounced and also visible in SI Fig.8 acquired with the O-LSBM. We agree with the Reviewer that a detailed comparison between a confocal and our LSBM would be informative, but this comparison should be done at the same wavelength (i.e. 780nm). Unfortunately, this is not technically feasible for us as we do not possess a 780nm confocal Brillouin microscope. However, we have added a detailed discussion towards this point to the new SI Note (*Comparison between point-scanning confocal and LSBM*).

16. What will be the results of the experiment of SI Fig. 9 using 780-nm confocal Brillouin imaging? Where will phototoxicity-optimized 780-nm confocal Brillouin microscopy appear in SI Fig. 4d? will it overlap the operation region of LSBM?

As noted in our reply above, such a direct comparison is unfortunately not technically feasible in our lab. But we can attempt to answer the second question by extrapolating from data in the published literature (e.g. Schluessler 2018, 2020, Antonacci 2020). As outlined in the SI Note (*Comparison between point-scanning confocal and LSBM*), our E-LSBM approach requires $\sim 500\times$ less illumination energy per pixel compared to a confocal Brillouin microscope implementation at 780nm (Schluessler 2018) which targeted comparable samples. Since the photodamage is proportional to the light dosage, this corresponds to an (at least) 500x lower photodamage compared to confocal Brillouin microscopy. Other demonstrations, notably Besner 2016 and Nikolic 2019 were either optimized for imaging of crystalline lens tissue which generates less-background than most biological specimen, or have utilized different wavelength, respectively.

We have also compiled a plot (SI Note Fig. 1) that shows where 780nm confocal BM implementations would fall on SI Fig. 4d. Indeed, it falls far outside (~ 100 -fold) of the operating region of LSBM!

Fig. R3: Precision vs. illumination energy for the orthogonal (red) and epi (blue) line geometries and comparable confocal Brillouin microscopy implementations. The fit shows a square root dependence, as expected in shot noise limited conditions. The shaded red region indicates the typical imaging conditions used in the experiments. Error bars represent S.D.

17. Can LSBM with an effective pixel time of 1 ms pixel be shown to achieve high-resolution images of living organisms comparable to those of confocal Brillouin microscopy (e.g., <https://doi.org/10.1364/BOE.10.001420>)?

Again, we refer to our replies above. We would furthermore like to again highlight that LSBM at 780 cannot achieve similar resolution to a 532nm confocal BM. The goal of our study was to design a low-phototoxicity BM approach that permits fast 3D BM imaging over extended time periods. We again refer to our extensive SI Note that discusses the performance differences of the various BM modalities.

18. What is the maximum frame rate of the LSBM instrument?

In principle the LSBM frame rate is limited by the EMCCD camera readout time (~10ms) and the stage movement for the scanning (few tens of ms) but in practice higher frame rates require higher illumination power to keep the same SNR and thus precision. We have added this discussion to the new SI note (*Acquisition time of the LSBM*).

19. If the analysis of the LSBM data comes down to the mean Brillouin frequency shift over the image (or over a few large, segmented regions of the image), why would not time lapsed imaging of a line in the sample by LSBM or multiple points in the sample by a 780-nm confocal Brillouin microscope with fluorescence guidance be adequate to study the samples presented in this work? This would enable to measure faster dynamics limited by the camera exposure time rather than by the volume imaging time of LSBM. Nevertheless, it should explicitly be noted in the manuscript that the fastest dynamics LSBM can probe is limited by the camera exposure time of ~100 ms rather than the effective pixel time.

While in principle we agree with the suggested strategy, we note that animal development, and in particular morphogenesis, involves by definition a dynamic, three-dimensional shape change of cells, tissues, organs and organisms as whole. Therefore, a Brillouin shift over a line (or set of points) would be an incomplete and potentially biased way to reflect the mechanical changes occurring during morphogenesis. Nevertheless, we think such a discussion is warranted and we now mention this possibility in the new SI note (*Acquisition time of the LSBM*), together with a discussion on limitations associated with the camera exposure time (which we also explicitly now state in the main manuscript).

20. Are 3 embryos an adequate sample size for the statistical analysis presented in Figs. 2d and 3j?

Both Ascidians and *Drosophila* embryos have in general a highly robust and reproducible development that generates little biological variability. Thus we believe a sample size of 3 should be representative of the species. Again, we highlight that the goal of our methods-oriented work was not to obtain statistically significant biological results, but to demonstrate the capability of LSBM to capture 3D mechanical properties of highly sensitive living organisms in biology which in turn opens the door to perform such analysis in principle.

Furthermore, we note that in our *Drosophila* experiments 3 embryos actually represent 3 independent experiments (different days, different batches of animals). Here, we have actually collected over the course of our study a much larger sample number ($N \sim 8$) in which 100% displayed the Brillouin shift trends we show in Fig. 2. The only reason why we have only reported 3 independent experiments is because only these 3 embryos were perfectly aligned in the FOV (Ventral midline at ~90°), which enabled the quantification of the Brillouin shift in the future mesoderm (the contractile domain, snail positive cells Fig. 2a).

We also note that Fig. 3h was now changed to include the tissue specific measurements of all three embryos (see also our response to point 11 above).

Reviewer #2:**Remarks to the Author:**

A Summary of Key Results: The Brillouin microscope reported here uses line scanning and epi or orthogonal detection to increase image speed to circa 1 ms per voxel whilst achieving up to 180x165x170 μ m field-of-view (FOV) with down to 1.5 μ m spatial and down to 2min temporal resolution, whilst achieving circa 80 dB spectral extinction and a spectral precision of <20 MHz. The microscope also incorporates a SPIM setup for co-registered fluorescence imaging.

The key results in this work are three examples to show that Brillouin microscopy's capability in relation to temporal dynamics (live drosophila embryo), high-mechanical property resolution (the ascidian *Phallusia mammillata*), and low photo-toxicity (mouse embryo). I would have found a table summarising the experiment, including the mode and variable FOV and spatial resolution to be very helpful in sorting out the myriad details. Could this be included?

We fully agree with this suggestion and have added an SI Table to the manuscript.

In epi mode at 20 mW illumination power is viable (no observed photodamage and 24-hr post viability) on live drosophila embryo over a time lapse of 30 minutes or so, at around 10 times lower illumination energy per pixel. Co-registered SPIM is included.

In orthogonal mode, the microscope was used to image the ascidian *Phallusia mammillata* over up to 14 hours – what was the illumination power of the laser? Co-registered SPIM is included.

The illumination power of the laser in this experiment was ~18mW in the sample. We have added all details in the respective sections and included a new SI Table. We also note that the experiment referenced by the Reviewer was actually recorded with the E-LSBM modality.

And (can the mode be stated please – paragraph commencing line 172) with acquisition times of ~11-17min at 75-90min intervals over 46 hours, sensitive mouse embryos were shown to be undamaged, whereas, damaged when imaged using "conventional" confocal imaging parameters and wavelength.

The mouse embryo imaging data shown in SI Fig. 8 was acquired in the orthogonal (O-LSBM) mode and chosen for its gentleness.

B Originality and significance: Line scanning in an orthogonal detection geometry was proposed and published in 2016 – Sci Reports: Ref 20. Can explicit reference be made to the novelty in the current manuscript? I expect this may be a collection of incremental improvements that collectively add up to a technically challenging and high-performance implementation – relative to previous incarnations, but this should be clearer from the manuscript. If the significance is to claim new mechanobiological capability, then this too should be clearly stated. Right now, it seems this manuscript would be well suited for Reviews of Scientific Instruments – a complex and highly capable instrument developed and described with exemplar samples and targets. The claimed advances need to be more specifically made and substantiated.

We carefully revised the manuscript to be more explicit about the novelty and conceptual advancements that our method enables. In short, we had to make significant improvements over the first line-scanning implementation of Ref. 20 (Zhang 2016) with respect to the achievable SNR, background suppression, spatial resolution, low photo-toxicity as well as the ability to maintain embryo viability during live imaging (see introduction section). All this combined allowed for the first time that mechanical properties of dynamic biological events were captured in 3D and over time at relatively high spatial resolution, without the need of invasive tissue

sectioning or particle injection (see new summary paragraph). We believe this demonstrates new mechanobiological capability.

In lines 189-191, in summary, it is claimed "Compared to alternative Brillouin scattering approaches and implementations^{7,9,13,14,18} this represents a >20-fold improvement in terms of imaging speed, at >10-fold lower illumination energy per pixel without sacrifices in measurement precision⁷." This is very clear, but it is not clear which specification is demonstrated by which experiment, i.e., whether they can be achieved simultaneously, and the claim of high mechanical property resolution of example 2, appears to be contradicted by the claim of without sacrificing measurement precision, when an improvement might have been expected.

We have clarified the summary statement mentioned by the Reviewer to better elaborate on the advancement being made. In short, both higher speed and lower illumination energy per pixel could indeed be obtained simultaneously, with similar measurement precision. We also note that the claim regarding measurement precision was made with respect to previous/standard work in the field (~10–20MHz, see Reviews Antonacci 2020, Prevedel 2019).

Overall, it is indeed encouraging to see Brillouin microscopy continue to progress as a non-contact mechanobiological imaging technique – I am supportive of publication, once minor issues are addressed and subject to a convincing explanation of novelty and significance.

We thank the Reviewer for their positive assessment and hope our revision succeeded in more convincingly explaining the novelty and significance!

C Data & methodology: There is a huge amount of detail in methods and supplementary information that broadly speaking is well presented and convincing. As indicated, a table summarising the three examples would help as there is a lot of change between experiments.

We fully agree with this suggestion and have added an SI Table to the manuscript.

Can the authors spell out any negative consequences of incorporating SPIM, for example, on photo-toxicity?

We have included a discussion towards this in the new SI Note (*Acquisition time of the LSBM*). In short, there are likely no negative effects as the SPIM imaging is performed only once every ~10-20min, i.e. for every Brillouin time-lapse volume, and the excitation powers are comparatively very low (<~0.1mW). But especially for very sensitive samples or embryos such as the mouse it can add to the overall photo-burden and thus needs to be taken into account when designing the imaging experiments (time-lapse interval etc.). At present, we believe there is no better alternative to SPIM for capturing fluorescence in terms of gentleness, i.e. low photo-toxicity, which is why we chose to incorporate this modality to our LSBM system.

For the orthogonal mode, it would seem that there is potential for droop in Brillouin frequency versus illumination depth, arising from refraction, sample heterogeneity, and the Brillouin frequency shift's dependence on angle to the incident beam. Was this observed? If so, what was its magnitude? If not, why not?

The Reviewer raises a good point which we realise warrants additional discussion. As already briefly discussed in the current manuscript, O-LSBM shares the same limitation as other orthogonal illumination & detection modalities such as SPIM. Here especially refraction effects and sample heterogeneity, as pointed out by the Reviewer, have to be considered and can potentially lead to artefact. In such cases, the geometry of the E-LSBM modality is better suited for more scattering or heterogeneous samples as it is inherently insensitive to these effects (as

already pointed out in the discussion). We now discuss these points in greater detail in a new SI Note (*Effect of refraction on the LSBM signal*) which we have added to the revised manuscript.

D Appropriate use of statistics: No specific comments.

E Conclusions: Overall, this manuscript provides a comprehensive description of the instrument and the experiments performed using it, but the wood is lost for some extent to the trees – the advances in capability are less clear. See above for novelty and significance. As regards the experimental findings, can the conclusion/summary be clear on what has been seen before and what is reported here for the first time.

We will take this feedback to heart and explain the novelty and significance more convincingly in the revised manuscript.

F Suggested improvements: experiments, data for possible revision: As above, greater clarity around what the novelty and significance is would help frame the manuscript as a pivotal one whereas at present this clarity is lost in the detail.

We have taken this feedback to heart and now explain the novelty and significance more convincingly in the revised manuscript.

G References: Appropriate credit is given to previous work.

H Clarity and context: I found the abstract rather weak and did not convey the key novelty and achievements well at all.

We have revised the abstract as best as we could. We note however that in the Brief Communication format the abstract is limited to 90 words only which prohibits extensive changes and more elaborate statements.

Reviewer #3:

Remarks to the Author:

The manuscript "High-resolution line-scan Brillouin microscopy for live-imaging of mechanical properties during embryo development" by Bevilacqua et al propose a new Brillouin microscopy design, which brings a much-needed improvement to the technique. Indeed, combining line-scanning approach with near infra-red illumination, their proposed design allows for a faster less-toxic acquisition of Brillouin microscopy images without sacrificing measurement precision, a quality that is important for long term imaging live samples and especially dynamic processes. Furthermore, the microscope has a dual mode: an O-LSBM to allow for better axial resolution and lower phototoxicity, and an E-LSBM that minimises effects from scattering and optical aberrations (at the cost of lower axial resolution) and thus is suited for heterogenous samples. Finally, adding a GPU-accelerated routine for data visualization makes the microscopy technique more practical and insightful to the user, as it allows for in situ visualization of the data. The authors test their microscope in three different model organisms, exploring the potential applications of their system.

The manuscript is written clearly, and the appropriate controls and measures are conducted in most places and described in detail. There are however a few comments and suggestions that I believe would improve the quality of the manuscript. Overall, I believe that this manuscript will open the way for increased use of Brillouin microscopy in biological/biomedical studies, and would therefore recommend publication in Nature Methods following consideration of the points below.

We thank the Reviewer for their overall positive assessment and their great suggestions! We have addressed most of the points as outlined below and believe these have further strengthened the manuscript. Some suggestions by the Reviewer have actually led to new results while others were not technically possible or unlikely to yield a meaningful outcome, as described below.

Main comments/suggestions:

Line 64-66: "Our microscope is based on a line-scanning approach that enables multiplexed signal acquisition, allowing the simultaneous sensing of hundreds of points and their spectra in parallel." Considering that 90° scattering geometry of LSBM broadens spectra and therefore limits resolution¹, could the authors please comment on if/how they have overcome this limit?

The Reviewer is correct that the 90deg scattering geometry broadens the resulting spectra, however, this does not necessarily impact spatial resolution (only spectral resolution). In our work, we only report on the measured Brillouin shifts, which are the peak positions of the fitted spectra, and those can still be obtained with relatively high precision (see Fig. 1f). In that sense, we did not overcome a limitation. In fact, in our case the higher spatial confinement of the illumination line in the O-LSBM, achieved through the synchronised use of an ETL lens, actually leads to an improved spatial resolution compared to the 180deg E-LSBM modality (see Fig. 1e).

E-LSBM vs O-LSBM: From Fig 1f, it seems that the Brillouin Shifts measured by the E-LSBM are systematically higher than those measured by O-LSBM. Is this something expected from the system (e.g. from the optical geometry/setup). Furthermore, Fig S14d shows that for the same illumination energy, the precision of O-LSBM is more than E-LSBM, hence being less toxic for live imaging. However this contradicts with Fig 1f, where it seems that although O-LSBM gives better spatial resolution, the precision of the E-LSBM is higher (smaller standard deviation). Could the authors please clarify this, for example perhaps different laser power/illumination energy was used when comparing the two modes in Fig 1f?

The Reviewer is correct that the Brillouin shifts in the E-LSBM are systematically higher, which is a direct consequence of the larger scattering angle (180 vs. 90 deg - the Brillouin shift scales with $\sin(\Theta)$). Furthermore, in SI Fig.4d indeed the precision of the O-LSBM has a higher value compared to the E-LSBM at the same laser power/illumination energy, which in fact means that the precision is less (or worse). Therefore, there is no contradiction and we believe this has been a misunderstanding of the Reviewer. We have added additional labels to SI Fig. 4d to make this point clearer in the revised manuscript.

Line 144-146: "Similar to the observations during VFF, the average Brillouin shift within cells engaged in tissue folding also increased during PM". The cells in the contractile region are probably those that move and deform most. I was wondering if the movement of cells could affect the light scattering and therefore the Brillouin shift? If so, then the increased Brillouin shift in this region might not necessarily be the result of the higher contractility but the result of higher movement? Could the authors please comment on this.

The Reviewer is correct that we observe higher Brillouin shifts when cells have started to undergo cell shape changes. However, cell movement and high Brillouin shift do not fully correlate: cells move in areas where we do not observe a high Brillouin shift. Consistently, we detect the transient high shift within cells that are ingressing, but not in cells moving on the from the sides towards the furrow. Furthermore, the apical sides of cells, which are compromised in the most dramatic cell shape change, apical constriction, does not show a high shift. Altogether, these data suggest the high Brillouin shift is not a consequence of cell movement but instead an evolving mechanical property of the tissue.

Additionally, from a physical perspective, moving cells could also in principle induce a Doppler shift on the scattered signal. However, practically this effect would be too small to be observable and also displace the Stokes and anti-Stokes in the same direction and therefore not change the detected Brillouin shift.

Line 147-149: "No photodamage or -toxicity was observed at $< \sim 20$ mW of average laser power, and viability assays showed that all embryos ($n=3$) imaged progressed to the first larval stage (24hpf)." The fact that all imaged embryos progressed to the first larval stage is indeed a strong indication of minimal photodamage. However, the authors could make their case even stronger if they showed that the imaged embryos developed into normal adults.

The common measure of survival after any treatment of a *Drosophila* embryo is reaching the first larval stage (L-1) which only depends on the fitness of the embryo itself. A larva that is able to hatch from the egg shell is considered to be fully viable. Using a later stage (beyond L-1) as suggested by the Reviewer would not be ideal, because not more than $\sim 80-90\%$ of L-1 survive to the adult stage in standard media, and larvae kept individually in vials are even less likely to survive (fly food humidity etc. is optimised for populations). In other words, survival beyond L-1 is affected by several additional stresses. Among these additional reasons are food accessibility due either to competition or lack of cooperation with other larvae, or failure to survive metamorphosis during pupariation. None of those would be related to the LSBM imaging, and therefore not informative about the potential photo-toxicity of our method. We hope the Reviewer finds these explanations acceptable.

Similarly, for ascidians, a common standard for good development is whether embryos are able to form normal larvae and not whether they form adult animals. The reasons for this are that although embryonic development is fairly fast, an embryo takes about 3 months to develop into an adult animal. Additionally, growing or maintaining animals bred in the laboratory has almost only been successfully achieved by laboratories located at marine stations with running sea

water. Therefore, following the suggestion of the Reviewer would be technically unachievable for Phallusia at EMBL in Heidelberg.

Line 157-160: "We observed a perinuclearly localized, high Brillouin signal within the B5.2 cells in the late 16-cell stage (Fig. 3e,f). This subcellular region is known to have a dense microtubule bundle structure driven by the centrosome attracting body (CAB) (Fig. 3d)." The authors use this example to show the ability of the O-LSBM to mechanically probe subcellular structures with high resolution, and this would suffice for the current manuscript. However, their result and manuscript would be much stronger if they could perturb the microtubule structure (for example with Nocodazole treatment) and show that the stiff region is gone, confirming that the high Brillouin shift region is indeed due to the dense microtubule bundle structure in these cells.

We fully agree with the Reviewer's that additional perturbation experiments would strengthen our observations. We thus set out to perform these experiments. In doing so we actually found that a strong peri-nuclearly localized signal is also present in other cell types, and not restricted to the germ-cell lineage alone as our data originally suggested. These new findings are summarised in the new SI Fig. 11.

As the Reviewer hypothesized, after nocodazole treatment the high Brillouin shift regions vanished in the embryos (see Fig. R4 below). However, due to the difficult nature of these perturbation experiments (low number of obtained embryos, precise timing and dose requirements as nocodazole stops the cell cycle), and despite our best efforts spanning several weeks we could only perform these perturbation experiments successfully in two embryos. Given the low number, we realize these experiments are not entirely conclusive and would therefore like to refrain from adding them to the paper. They are however attached below for the Reviewer. As a result, we decided to remove any statements specifically related to the germ cells and or microtubule bundles from the manuscript and instead state that further experiments are needed to pinpoint the origin of the peri-nuclear signal.

Fig. R4: (a,b) Orthogonal view of two nocodazole treated embryos. Brillouin shift map (top) and SPIM images of membranes (labelled with FM4-64), confirming the 16-cell stage characteristic morphology. The embryos were treated with nocodazole (2.6μM in sea water) from early 16-cell stage before being imaged at late 16-cell stage in the presence of nocodazole. As the nocodazole treatment causes a developmental arrest by preventing cell division, developmental progression of nocodazole treated embryos was assessed by the developmental progression of synchronously developing non-treated sibling embryos.

Fig 3f: It seems that the authors have Brillouin microscopy images of 2 organisms (caption of figure 3b,c). It would therefore be beneficial to plot the Brillouin shift along a similar line in the 2nd organism to show reproducibility.

Please see our response above. We have added additional images as a separate, new SI Fig. 11.

Line 167-169: "These experiments also demonstrate that LSBM imaging can be used over long periods (here 14 hours; SI Video 3-4)". The authors indeed show that the organism is viable over long period of imaging. However, a more quantitative analysis (e.g. LIVE/DEAD assay or similar methods) would be desirable, especially because low phototoxicity is one of the main features of the proposed microscopy method.

We agree that a more quantitative analysis would be informative and have performed additional LIVE/DEAD assays, i.e. we subjected another 3 ascidian embryos to long-term LSBM imaging. Again, 100% of these survived, which raises the overall number to n=6 in two independent experiments, which we now report in the manuscript.

Line 172-174: "Finally, to test the low-photo-toxicity of the LSBM approach in a further organism, we imaged the developing mouse embryo, from the 8-cell stage (E2.75) to the late-blastocyst (E4.5), covering a 46-hour time-span (SI Fig. 8)." The authors only show the initial and final timepoints of this experiment. It would be beneficial to include a movie of this process so the reader could see the intermediate timepoints as well.

We have included new SI videos 5 and 6, showing maximum intensity projection and single plane movies over time, respectively, in the revised version.

Line 177-180: "Despite the embryos' notorious photo-sensitivity, no photodamage or -toxicity was observed at $< \sim 20$ mW of average laser power as confirmed by the morphology, dynamics, cell number and cell fate of the imaged embryos resembling those of control embryos (SI Fig. 8c,d)." Currently the authors show exemplary images of the imaged embryos and show that they are normal in terms of morphology and cell fate. It would be beneficial to include more quantitative measures of cell number and dynamics to confirm that the long-term imaging was non-toxic.

We have added the requested details on cell numbers to the manuscript (SI Fig. 8c,d), and the new SI videos 5,6 show size and cavity oscillation dynamics which are indicative of proper development. We believe this gives a more quantitative proof of the non-phototoxicity of our method.

Minor comments:

We thank the Reviewer for spotting the below typos and missing details!

Line 324: Typo. "raise" should be "rise".

We have corrected this.

Fig 1e: Do the circle, cross and star represent different experiments? Also, for both Fig 1e and f, it would be beneficial to include a more detailed description of the measurements and how they should be interpreted in the caption.

We have expanded the caption and Methods section on how the measurements were done.

Fig 3j: What do the inner error bars represent?

We have changed the content of this panel during the revision.

Fig S4 title: Typo. "BLSM" should probably be "LSBM" to be consistent with the rest of the manuscript.

We have corrected this.

Fig S4d, S7a: Please mention what the error bars represent.

We have added this detail to the Figure caption.

Fig S8c,d: Is the n number the same as Fig S8a, i.e. n=4?

We have added this detail to the Figure caption. The number is n=3 because 1 embryo got lost during the staining procedure

Fig S9b,d: Numbers on the colorbar are hard to read.

We have corrected this.

References:

1 Prevedel, R., Diz-Muñoz, A., Ruocco, G. & Antonacci, G. Brillouin microscopy: an emerging tool for mechanobiology. *Nature Methods* 16, 969-977, doi:10.1038/s41592-019-0543-3 (2019).

Decision Letter, first revision:

Dear Robert,

Thank you for submitting your revised manuscript "High-resolution line-scan Brillouin microscopy for live-imaging of mechanical properties during embryo development" (NMETH-BC48320A) and for your patience during the review process. It has now been seen by the original referees and their comments are below. The reviewers find that the paper has improved in revision, and therefore we'll be happy in principle to publish it in Nature Methods, pending minor revisions to satisfy the referees' final requests and to comply with our editorial and formatting guidelines.

Specifically, please make sure that any limitations are discussed in the main text.

TRANSPARENT PEER REVIEW

Nature Methods offers a transparent peer review option for new original research manuscripts submitted from 17th February 2021. We encourage increased transparency in peer review by publishing the reviewer comments, author rebuttal letters and editorial decision letters if the authors agree. Such peer review material is made available as a supplementary peer review file. Please state in the cover letter 'I wish to participate in transparent peer review' if you want to opt in, or 'I do not wish to participate in transparent peer review' if you don't. Failure to state your preference will result in delays in accepting your manuscript for publication.

ORCID

IMPORTANT: Non-corresponding authors do not have to link their ORCIDs but are encouraged to do so. Please note that it will not be possible to add/modify ORCIDs at proof. Thus, please let your co-authors know that if they wish to have their ORCID added to the paper they must follow the procedure

described in the following link prior to acceptance:

Best regards,
Nina

Nina Vogt, PhD
Senior Editor
Nature Methods

Reviewer #1 (Remarks to the Author):

In the revision, the authors addressed adequately the issue of sample photodamage, but addressed only to some extent my concerns regarding the spatial/spectral/mechanical resolution, precision, acquisition time, and fluorescence guidance in LSBM. It appears that the present LSBM system with 0.8-NA illumination and detection objectives and ~1-ms effective pixel time is appropriate for longitudinal imaging experiments in large samples but with limited 3D resolution and spatial averaging of the Brillouin shift, as discussed mainly in the supplementary material. I think that the main points of this discussion need to appear in the manuscript.

It is still unclear to me whether the cost for achieving sufficiently fast imaging speed in the selected samples was at the expense of a good mechanical contrast (and resolution), which looks weak (and blurred) in all the images and too pixelated in the zoomed images. No experimental evidence was provided on the effectiveness of LSBM for fast mechanical imaging with sub-micrometer resolution, which is important in many biological studies (e.g., cells). Thus, I think that claims about high resolution are misleading. Also, the statement about the benefit of resolving closely spaced Brillouin peaks in the probed volume is not fully accurate (see 10.1038/lsa.2017.139). The Editor will evaluate the fit of the manuscript to this high profile, broad interest journal.

Minor comments:

1. In the supplementary material it is stated that “We designed the thickness of the illumination line to be ~1 μ m, that is the typical mechanical resolution achievable in a biological sample.”, where in the

rebuttal letter it is written “We designed the size of the illumination line to be below $\sim 2\mu\text{m}$, that is the typical mechanical resolution in biological samples.” Which statement is more correct?

2. The expression $\lambda_a = V \cdot \Gamma_B^{-1}$ in the supplementary material should be corrected to $\lambda_a = V \cdot (\Gamma_B / 2\pi)^{-1}$
3. The results in Figure 3h are confusing and a biological interpretation of these results would be helpful.
4. Cannot the E-LSBM path be slightly modified to provide also CBM at 780 nm?
5. 3D render images of the data would be a valuable addition.

Reviewer #2 (Remarks to the Author):

I observe that the authors have rigorously and comprehensively addressed the comments of three reviewers - my own to my complete satisfaction. In particular, the novelty has been clarified to my satisfaction. Extensive materials have been added to the Supplement which help clarify many details. My only advice would be to consider the number of significant figures in places - for example, in Figs 1e and 1f, are four significant figures really justified? I would ask the authors to review and consider this, here, and elsewhere.

Reviewer #3 (Remarks to the Author):

The revised manuscript has been substantially improved with new supporting data/analyses that further clarify the method and biological conclusions. Sufficient major reviewers' concerns have been addressed, and we therefore recommend the paper for publication.

Author Rebuttal, first revision:

Point-by-point reply for NMETH-BC48320**Remark to all Reviewers:**

We would like to thank all Reviewers for their reports and valuable comments, which we believe, have helped to significantly enhance the quality of our manuscript. Below we give a point-by-point response to all issues that were raised and how we have addressed them in the revised version of our manuscript. To facilitate review of the revised manuscript, we have underlined any additions to the text or areas with other significant changes. We are confident that these improvements should fully address all Reviewers' comments and suggestions, and hope that it now meets their expectations.

Original Reviewer comments are in black. **Our replies are in blue.** *Changes to manuscript text are included in red where appropriate.*

Reviewer #1 (Remarks to the Author):

In the revision, the authors addressed adequately the issue of sample photodamage, but addressed only to some extent my concerns regarding the spatial/spectral/mechanical resolution, precision, acquisition time, and fluorescence guidance in LSBM. It appears that the present LSBM system with 0.8-NA illumination and detection objectives and ~1-ms effective pixel time is appropriate for longitudinal imaging experiments in large samples but with limited 3D resolution and spatial averaging of the Brillouin shift, as discussed mainly in the supplementary material. I think that the main points of this discussion need to appear in the manuscript.

We thank the Reviewer for this suggestion. We note that we are already discussing the pros and cons of our LSBM in the Discussion section, but have now added a concrete note about the 3D spatial resolution limitation to the main manuscript. In particular, we now state in the Discussion section:

We note that while the spatial resolution of our LSBM is substantially higher than previous line-scan BM implementations²⁰, it is lower than confocal BM^{9,14}.

It is still unclear to me whether the cost for achieving sufficiently fast imaging speed in the selected samples was at the expense of a good mechanical contrast (and resolution), which looks weak (and blurred) in all the images and too pixelated in the zoomed images. No experimental evidence was provided on the effectiveness of LSBM for fast mechanical imaging with sub-micrometer resolution, which is important in many biological studies (e.g., cells). Thus, I think that claims about high resolution are misleading. Also, the statement about the benefit of resolving closely spaced Brillouin peaks in the probed volume is not fully accurate (see 10.1038/lsa.2017.139). The Editor will evaluate the fit of the manuscript to this high profile, broad interest journal.

We apologize if our careful revision left some aspects this unclear. However, we respectfully disagree with some of the Reviewer's assessments.

With respect to the mechanical contrast (and resolution) looking weak and blurred: Fig. 1e,f shows clear quantifications of our spatial resolution and precision, which are on-par with other state-of-the-art Brillouin microscopes. The impression might be partly due to the fact that we use a linear color scale (MPL-Inferno in Fiji) to represent our Brillouin shift data, which gives visually 'less striking' images. However, their use is highly recommended in order to prevent the accentuation of small differences.

Furthermore, we note that we have never claimed to achieve *sub-micrometer* resolution as suggested by the Reviewer. Again, our spatial resolution is quantified in Fig. 1e,f and in fact, we claim *sub-cellular* resolution, which we believe is fairly evident from our data, eg. Fig. 3a,c.

Finally, we highlight that the spatial resolution was not meant nor designed to be as good as in a confocal BM implementation. The claims about high-resolution are clearly made with respect, and in comparison, to previous line-scan implementations (see Ref. 20 - which used NA=0.1 only, compared to NA=0.8 in our work.). Here, we also refer to the added note in the Discussion section (see Response above), which we hope will further clarify this point in the paper.

Lastly, with respect to our statement in the SI Note 1 about the benefit of resolving close Brillouin peaks, we have reworded the statement in order to be less ambiguous:

We note, however, that even though resolving different peaks might provide additional information about the heterogeneity of the sample within the probed volume (PSF), this information is not improving the spatial resolution per se.

Minor comments:

1. In the supplementary material it is stated that "We designed the thickness of the illumination line to be $\sim 1\mu\text{m}$, that is the typical mechanical resolution achievable in a biological sample.", where in the rebuttal letter it is written "We designed the size of the illumination line to be below $\sim 2\mu\text{m}$, that is the typical mechanical resolution in biological samples." Which statement is more correct?

We thank the Reviewer for pointing out this inconsistency. This has been indeed a typo in the rebuttal letter and the information in the Supplementary Material is correct.

2. The expression $L_a = V \Gamma B^{-1}$ in the supplementary material should be corrected to $L_a = V (\Gamma B / 2\pi)^{-1}$

We thank the Reviewer for spotting this and have corrected our definition of the linewidth and adjusted the expression for L_a accordingly in the revised version.

3. The results in Figure 3h are confusing and a biological interpretation of these results would be helpful.

The results in Figure 3h show the analysis of the mechanical properties within particular cell populations of *Phallusia* embryos: epidermal, central nervous system and endo-mesodermal fates. The acquisition of particular fates is driven by dynamic genetic expression programs that may have an impact on the behaviours and mechanical properties of cells. For example, cells committed to acquire mesodermal fate in *Drosophila* embryos generate actomyosin contractility downstream of the pro-mesodermal fate determinants Snail and Twist. Among Ascidians, *Ciona*, which has reported high conservation of gene expression patterns with *Phallusia* (Madgwick 2019, PMID: 30661644), displays differential CadherinII expression in the endoderm and the sensory vesicle (Noda & Satoh 2008, PMID: 18400563), a cell population of the CNS. Thus, it is possible that the different cell populations of *Phallusia* embryos acquire particular mechanical properties as a consequence of their respective differentiation programs. However, it was not possible for us to predict based on the current knowledge of *Phallusia* development, how the mechanical properties of these embryonic cell populations could differ. Therefore, we quantified the Brillouin shift in these cell populations across 3 embryos measured in independent experiments. We found that the mechanical properties of these embryonic cell populations do not significantly differ (as per the statistical analysis). However, in all embryos (3/3), we found a trend, and that is, the endo-mesodermal population is stiffer than the rest of cell populations in the embryo. We might further explore this direction in the near future, and assess whether

Cadherin11 expression pattern overlays the regions of the Phallusia embryo that display high Brillouin shifts, which could establish for the first time a connection between cell adhesion properties and measured Brillouin shifts at the organismal level.

4. Cannot the E-LSBM path be slightly modified to provide also CBM at 780 nm?

We thank the Reviewer for this suggestion. While indeed the LSBM path could in principle be modified to enable confocal BM measurements, it practically requires the removal (and thus the realignment) of the cylindrical lens and the substitution of the confocal slit with a pinhole (with the same consideration on alignment). A switchable configuration to be used on the same sample would be technically challenging and require redesign of the current optical mount of the cylindrical lens and slit.

5. 3D render images of the data would be a valuable addition.

We agree with the Reviewer that a 3D rendering would be enticing, however we note that unlike in fluorescence imaging, in BM every image voxel contains information (e.g. a frequency shift value). Therefore a 3D rendering of a particular object is not straightforward without considerable thresholding or otherwise choosing of a (narrow) frequency window. Such postprocessing would therefore yield a very biased representation of the 3D image and likely not be very informative. We have therefore refrained from implementing this request.

Reviewer #2 (Remarks to the Author):

I observe that the authors have rigorously and comprehensively addressed the comments of three reviewers - my own to my complete satisfaction. In particular, the novelty has been clarified to my satisfaction. Extensive materials have been added to the Supplement which help clarify many details. My only advice would be to consider the number of significant figures in places - for example, in Figs 1e and 1f, are four significant figures really justified? I would ask the authors to review and consider this, here, and elsewhere.

We thank the Reviewer for their positive comments and assessment. Concerning the number of significant figures in Fig. 1e we determined them from the error bars in SI Fig. 4d while in Fig. 1f we kept the first digit after the comma, compatible with the step size used for acquiring the PSF.

Reviewer #3 (Remarks to the Author):

The revised manuscript has been substantially improved with new supporting data/analyses that further clarify the method and biological conclusions. Sufficient major reviewers' concerns have been addressed, and we therefore recommend the paper for publication.

We thank the Reviewer for their positive assessment of our work!

Final Decision Letter:

Dear Robert,

I am pleased to inform you that your Article, "High-resolution line-scan Brillouin microscopy for live-imaging of mechanical properties during embryo development", has now been accepted for publication in Nature Methods. Your paper is tentatively scheduled for publication in our May print issue, and will be published online prior to that. The received and accepted dates will be February 9th, 2022 and February 17th, 2023. This note is intended to let you know what to expect from us over the next month or so, and to let you know where to address any further questions.

Once your paper is typeset, you will receive an email with a link to choose the appropriate publishing options for your paper and our Author Services team will be in touch regarding any additional information that may be required.

Please note that *Nature Methods* is a Transformative Journal (TJ). Authors may publish their research with us through the traditional subscription access route or make their paper immediately open access through payment of an article-processing charge (APC). Authors will not be required to make a final decision about access to their article until it has been accepted. [Find out more about Transformative Journals](https://www.springernature.com/gp/open-research/transformative-journals)

Your paper will now be copyedited to ensure that it conforms to Nature Methods style. Once proofs are generated, they will be sent to you electronically and you will be asked to send a corrected version within 24 hours. It is extremely important that you let us know now whether you will be difficult to contact over the next month. If this is the case, we ask that you send us the contact information (email, phone and fax) of someone who will be able to check the proofs and deal with any last-minute problems.

If, when you receive your proof, you cannot meet the deadline, please inform us at rjsproduction@springernature.com immediately.

Once your manuscript is typeset and you have completed the appropriate grant of rights, you will receive a link to your electronic proof via email with a request to make any corrections within 48 hours. If, when you receive your proof, you cannot meet this deadline, please inform us at rjsproduction@springernature.com immediately.

Once your paper has been scheduled for online publication, the Nature press office will be in touch to confirm the details.

Once your paper has been scheduled for online publication, the Nature press office will be in touch to confirm the details.

Content is published online weekly on Mondays and Thursdays, and the embargo is set at 16:00 London time (GMT)/11:00 am US Eastern time (EST) on the day of publication. If you need to know the exact publication date or when the news embargo will be lifted, please contact our press office after you have submitted your proof corrections. Now is the time to inform your Public Relations or Press Office about your paper, as they might be interested in promoting its publication. This will allow them time to prepare an accurate and satisfactory press release. Include your manuscript tracking number NMETH-A48320B and the name of the journal, which they will need when they contact our office.

About one week before your paper is published online, we shall be distributing a press release to news organizations worldwide, which may include details of your work. We are happy for your institution or

funding agency to prepare its own press release, but it must mention the embargo date and Nature Methods. Our Press Office will contact you closer to the time of publication, but if you or your Press Office have any inquiries in the meantime, please contact press@nature.com.

Nature Portfolio journals [encourage authors to share their step-by-step experimental protocols](https://www.nature.com/nature-research/editorial-policies/reporting-standards#protocols) on a protocol sharing platform of their choice. Nature Portfolio 's Protocol Exchange is a free-to-use and open resource for protocols; protocols deposited in Protocol Exchange are citable and can be linked from the published article. More details can found at www.nature.com/protocolexchange/about.

Please note that you and any of your coauthors will be able to order reprints and single copies of the issue containing your article through Nature Portfolio 's reprint website, which is located at <http://www.nature.com/reprints/author-reprints.html>. If there are any questions about reprints please send an email to author-reprints@nature.com and someone will assist you.

Best regards,
Nina

Nina Vogt, PhD
Senior Editor
Nature Methods